# PRUNEFUSE: EFFICIENT DATA SELECTION VIA WEIGHT PRUNING AND NETWORK FUSION

## ABSTRACT

Efficient data selection is crucial for enhancing the training efficiency of deep neural networks and minimizing annotation requirements. Traditional methods often face high computational costs, limiting their scalability and practical use. We introduce PruneFuse, a novel strategy that leverages pruned networks for data selection and later fuses them with the original network to optimize training. PruneFuse operates in two stages: First, it applies structured pruning to create a smaller pruned network that, due to its structural coherence with the original network, is well-suited for the data selection task. This small network is then trained and selects the most informative samples from the dataset. Second, the trained pruned network is seamlessly fused with the original network. This integration leverages the insights gained during the training of the pruned network to facilitate the learning process of the fused network while leaving room for the network to discover more robust solutions. Extensive experimentation on various datasets demonstrates that PruneFuse significantly reduces computational costs for data selection, achieves better performance than baselines, and accelerates the overall training process.

## 1 INTRODUCTION

Deep learning models have achieved remarkable success across various domains, ranging from image recognition to natural language processing (Ren et al., 2015; Long et al., 2015; He et al., 2016). However, the performance of models heavily relies on the access of large amounts of labeled data for training (Sun et al., 2017). In practical real-world applications, the process of manually annotating massive datasets can be prohibitively expensive and time-consuming. Data selection techniques such as Active Learning (Gal et al., 2017) offer a promising solution to address this challenge by iteratively selecting the most informative samples from the unlabeled dataset for annotation. The goal of active learning is to reduce the labeling costs while maintaining or even improving model performance. Nowadays, due to tremendous increase in data and model complexity, traditional active learning techniques requiring large models to be trained iteratively to perform data selection, can result in significant computational costs. This computational burden restricts the scalability of active learning methods, particularly in scenarios where training large models is impractical due to resource constraints.

In this paper, we propose a novel strategy for efficient data selection in active learning setting that overcomes the limitations of traditional approaches. Our approach builds up on the concept of model pruning, which selectively reduces the complexity of neural networks while preserving their accuracy. By utilizing small pruned networks as reusable data selectors, we eliminate the need to train large models, specifically during the data selection phase, thus significantly reducing computational demands. By enabling swift identification of the most informative samples, our method not only enhances the efficiency of active learning but also ensures its scalability and cost-effectiveness in resource-limited settings. Additionally, we employ these pruned networks to train the final model through a fusion process, effectively harnessing the insights from the trained networks to accelerate convergence and improve the generalization of the final model.

**Main Contribution.** To summarize, our key contribution is to introduce PruneFuse, an efficient and rapid data selection technique that leverages pruned networks. This approach mitigates the need for continuous large model training prior to data selection, which is inherent in conventional active learn-

Figure 1: **Overview of the PruneFuse Method**: (1) An untrained neural network is initially pruned to form a structured, pruned network $\theta_p$. (2) This pruned network $\theta_p$ queries the dataset to select prime candidates for annotation, similar to active learning techniques. (3) $\theta_p$ is then trained on these labeled samples to form the trained pruned network $\theta_p^*$. (4) The trained pruned network $\theta_p^*$ is fused with the base model $\theta$, resulting in a fused model. (5) The fused model is further trained on a selected subset of the data, incorporating knowledge distillation from $\theta_p^*$. Blue feedback indicates the PruneFuse V2 strategy deliniated in Section 4.6 that utilizes the trained fused model to create the pruned model.

ing methods. By employing pruned networks as data selectors, PruneFuse ensures computationally efficient selection of informative samples which leads to overall superior generalization. Furthermore, we propose the novel concept of fusing these pruned networks with the original untrained model, enhancing model initialization and accelerating convergence during training.

We demonstrate the broad applicability of PruneFuse across various network architectures, providing researchers and practitioners with a flexible tool for efficient data selection in diverse deep learning settings. Extensive experimentation on CIFAR-10, CIFAR-100, Tiny-ImageNet-200, and ImageNet-1K datasets shows that PruneFuse achieves superior performance to state-of-the-art active learning methods while significantly reducing computational costs.

## 2 RELATED WORKS

**Data Selection.** Recent studies have explored techniques to improve the efficiency of data selection in deep learning. Approaches such as Core-Set selection (Sener and Savarese, 2017), BatchBALD (Kirsch et al., 2019), and Deep Bayesian Active Learning (Gal et al., 2017) aim to select informative samples using techniques like diversity maximization and Bayesian uncertainty estimation. Parallelly, the domain of active learning has unveiled strategies, such as uncertainty sampling (Shen et al., 2017; Sener and Savarese, 2018; Kirsch et al., 2019), expected model change-based approach (Freytag et al., 2014; Käding et al., 2016), and query-by-density (Sener and Savarese, 2017). These techniques prioritize samples that can maximize information gain, thereby enhancing model performance with minimal labeling effort. While these methods achieve efficient data selection, they still require training large models for the selection process, resulting in significant computational overhead. Other strategies such as (Killamsetty et al., 2021a) optimize this selection process by matching the gradients of subset with training or validation set based on orthogonal matching algorithm and (Killamsetty et al., 2021b) performs meta-learning based approach for online data selection. SubSelNet (Jain et al., 2024) proposes to approximate a model that can be used to select the subset for various architectures without retraining the target model, hence reducing the overall overhead. However, it involves pre-training routine which is very costly and needed again for any change in data or model distribution. SVP (Coleman et al., 2019) introduces to use small proxy models for data selection but discards these proxies before training the target model. Additionally, structural discrepancies between the proxy and target models may result in sub-optimal data selections. Our approach also builds on this foundation of using small model (which in our case is a pruned model) but it enables direct integration with the target model through the fusion process. This ensures that the knowledge acquired during data selection is retained and actively contributes to the training of the original model. Also, the architectural coherence between the pruned and the target model provides a more seamless and effective mechanism for data selection, enhancing overall model performance and efficiency.

**Efficient Deep Learning.** Efficient deep learning has gained significant attention in recent years. Methods such as Neural Architecture Search (NAS) (Zoph and Le, 2016; Wan et al., 2020), network pruning (Han et al., 2015), quantization (Dong et al., 2020; Jacob et al., 2018; Zhou et al., 2016), and knowledge distillation (Hinton et al., 2015; Yin et al., 2020) have been proposed to reduce model size and computational requirements. Neural Network pruning has been extensively investigated as a technique to reduce the complexity of deep neural networks (Han et al., 2015). Pruning strategies can be broadly divided into Unstructured Pruning (Dong et al., 2017; Guo et al., 2016; Park et al., 2020) and Structured Pruning (Li et al., 2016; He et al., 2017; You et al., 2019; Ding et al., 2019) based on the granularity and regularity of the pruning scheme. Unstructured pruning often yields a superior accuracy-size trade-off whereas structured pruning offers practical speedup and compression without necessitating specialized hardware. While pruning literature suggests pruning after training (Renda et al., 2020) or during training (Zhu and Gupta, 2017; Gale et al., 2019), recent research explore the viability of pruning at initialization (Lee et al., 2018; Frankle et al., 2020; Tanaka et al., 2020; Frankle et al., 2020; Wang et al., 2020). In our work, we leverage the benefits of model pruning at initialization to create a small representative model for efficient data selection, allowing for the rapid identification of informative samples while minimizing computational requirements.

## 3 BACKGROUND AND MOTIVATION

Efficient data selection is paramount in modern machine learning applications, especially when dealing with deep neural networks. The subset selection problem can be framed as the challenge of selecting a subset $s$ from a dataset $D = (x_i, y_i)_{i=1}^n$ such that a model $\theta$ trained on $s$ approximates the performance of the same model trained on the full dataset,

$$\arg\min_s \big| E_{(x,y)\in s}[l(x,y;\theta)] - E_{(x,y)\in D}[l(x,y;\theta)] \big| \tag{1}$$

Where $E_{(x,y)\in s}[l(x,y;\theta)]$ is the expected loss on the selected subset $s$ and $E_{(x,y)\in D}[l(x,y;\theta)]$ is the expected loss on whole dataset.

### 3.1 SUBSET SELECTION FRAMEWORK

Active Learning is widely utilized iterative approach tailored for situations with abundant unlabeled data. Given a classification task with $C$ classes and a large pool of unlabeled samples $U$, AL revolves around selectively querying the most informative samples from $U$ for labeling. The process commences with an initial set of randomly sampled data $s^0$ from $U$, which is subsequently labeled. In subsequent rounds, AL augments the labeled set $L$ by adding newly identified informative samples. This cycle repeats until a predefined number of labeled samples $b$ are selected.

### 3.2 NETWORK PRUNING AND ITS RELEVANCE

Network pruning emerges as a potent tool to reduce the complexity of neural networks. By eliminating redundant parameters, pruning preserves vital network functionalities while streamlining its architecture. Pruning strategies can be broadly categorized into Unstructured Pruning and Structured Pruning. Unstructured Pruning targets individual weight removal independent of their location. While it trims down the overall number of parameters, tangible performance gains on conventional hardware often demand extensive pruning (Park et al., 2016). On the other hand, Structured Pruning emphasizes the removal of larger constructs like kernels, channels, or layers. Its strength lies in preserving dense computations, which not only yields a leaner network but also bestows immediate performance improvements (Liu et al., 2017). Given its computational benefits, particularly in expediting evaluations and aligning with hardware optimizations, we opted for Structured Pruning over its counterpart.

Importantly, pruned networks maintain the architectural coherence of the original model. This coherence makes them inherently more suitable for tasks such as data selection. Unlike heavily modified or entirely different models that can be used for data selection Coleman et al. (2019); Jain et al. (2024), the pruned model echoes the original structure, particularly advantageous in recognizing and prioritizing data samples that resonate with the patterns of the original network. The goal is clear to develop a data selection strategy that conserves computational resources, minimizes memory overhead, and potentially improves model generalization.

## 4 PRUNEFUSE

In this section, we delineate the PruneFuse methodology. The procedure begins with network pruning at initialization, offering a streamlined model for data selection. Upon attaining the desired data subset, the pruned model undergoes a fusion process with the original network, leveraging the structural coherence between them. The fused model is subsequently refined through knowledge distillation, enhancing its performance. An overall view of our proposed methodology is illustrated in Fig. 1. We modify the problem in Eq. 1 as follows:

Let $s_p$ be the subset selected using a pruned model $\theta_p$ and $s$ be the subset selected using the original model $\theta$. We want to minimize:

$$\arg\min_{s_p} \left| E_{(x,y)\in s_p}[l(x, y; \theta, \theta_p)] - E_{(x,y)\in D}[l(x, y; \theta)] \right| \tag{2}$$

Where $E_{(x,y)\in s_p}[l(x, y; \theta, \theta_p)]$ is the expected loss on subset $s_p$ (selected using $\theta_p$) when evaluated using the original model $\theta$ and $E_{(x,y)\in D}[l(x, y; \theta)]$ is the expected loss on full dataset $D$ when trained using the original model $\theta$. Furthermore, the subset can be defined as $s_p = \{(x_i, y_i) \in D \mid \text{score}(x_i, y_i; \theta_p) \geq \tau\}$ where $\text{score}(x_i, y_i; \theta_p)$ represents the score assigned to each sample selected using $\theta_p$. The score function can be based on various strategies such as Least Confidence, Entropy, or Greedy k-centers. $\tau$ defines the threshold used in the score-based selection methods (Least Confidence or Entropy) to determine the inclusion of a sample in $s_p$.

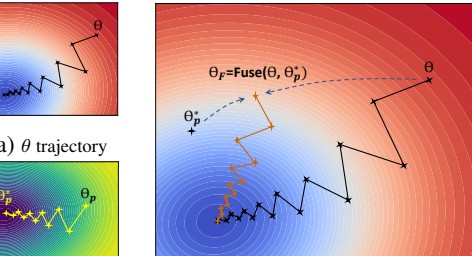

(a) $\theta$ trajectory

(b) $\theta_p$ trajectory     (c) $\theta_F$ with a refined trajectory due to fusion

Figure 2: **Evolution of training trajectories**. Pruning $\theta$ to $\theta_p$ tailors the loss landscape from 2a to 2b, allowing $\theta_p$ to converge on an optimal configuration, denoted as $\theta_p^*$. This model, $\theta_p^*$, is later fused with the original $\theta$, which provides better initialization and offer superior trajectory for $\theta_F$ to follow, as depicted in 2c.

The goal of the optimization problem is to select $s_p$ such that when $\theta$ is trained on it, the performance is as close as possible to training $\theta$ on the full dataset $D$. Algorithm 1 describes the PruneFuse methodology precisely. The key insight is that the subset $s_p$ selected using the pruned model $\theta_p$ is sufficiently representative and informative for training the original model $\theta$. This is because $\theta_p$ maintains a structure that is essentially identical to $\theta$, although with some nodes pruned. As a result, there is a strong correlation between $\theta$ and $\theta_p$, ensuring that the selection made by $\theta_p$ effectively minimizes the loss when $\theta$ is trained on $s_p$. By leveraging this surrogate $\theta_p$, which is both computationally efficient and structurally coherent with $\theta$, we can select most representative data out of $D$ to train $\theta$.

### 4.1 PRUNING AT INITIALIZATION

Pruning at initialization has been demonstrated to uncover superior solutions compared to the conventional approach of pruning an already trained network followed by fine-tuning (Wang et al., 2020). Specifically, it shows potential in training time reduction, and enhanced model generalization. In our methodology, we employ structured pruning due to its benefits such as maintaining the architectural coherence of the network, enabling more predictable resource savings, and often leading to better-compressed models in practice.

Consider an untrained neural network, represented as $\theta$. Let each layer $\ell$ of this network have feature maps or channels denoted by $c^\ell$, with $\ell \in \{1, \ldots, L\}$. Channel pruning results in binary masks $m^\ell \in \{0, 1\}^{d^\ell}$ for every layer, where $d^\ell$ represents the total number of channels in layer $\ell$. The pruned subnetwork, $\theta_p$, retains channels described by $c^\ell \odot m^\ell$, where $\odot$ symbolizes the element-wise product. The sparsity $p \in [0, 1]$ of the subnetwork illustrates the proportion of channels that are pruned: $p = 1 - \sum_\ell m^\ell / \sum_\ell d^\ell$.

To reduce the model complexity, we employ channel pruning procedure $prune(C, p)$. This prunes to a sparsity $p$ via two primary functions: i) score(C): This operation assigns scores $z^\ell \in \mathbb{R}^{d^\ell}$ to every channel in the network contingent on their magnitude (using the L2 norm). The channels

$C$ are represented as $(c_1, \ldots, c_L)$. and ii) remove(Z, p): This process takes the magnitude scores $Z = (z_1, \ldots, z_L)$ and translates them into masks $m^\ell$ such that the cumulative sparsity of the network, in terms of channels, is $p$. We employ a one-shot channel pruning that scores all the channels simultaneously based on their magnitude and prunes the network from 0% sparsity to $p$% sparsity in one cohesive step. Although previous works suggest re-initializing the network to ensure proper variance (van Amersfoort et al., 2020). However, since the performance increment is marginal, we retain the weights of the pruned network before training.

## 4.2 Data Selection via Pruned Model

We begin by randomly selecting a small subset of data samples, denoted as $s^0$, from the unlabeled pool $U = \{x_i\}_{i \in [n]}$ where $[n] = \{1, ..., n\}$. These samples are then annotated. The pruned model $\theta_p$ is trained on this labeled subset $s^0$, resulting in the trained pruned model $\theta_p^*$. With $\theta_p^*$ as our tool, we venture into the larger unlabeled dataset $U$ to identify samples that are prime candidates for annotation.

Regardless of the scenario, our method employs three distinct criteria for data selection: Least Confidence (LC) (Settles, 2012), Entropy (Shannon, 1948), and Greedy k-centers (Sener and Savarese, 2017). Least Confidence based selection gravitates towards samples where the pruned model exhibits the least confidence in its predictions. The confidence score is essentially the highest probability the model assigns to any class label. Thus, the uncertainty score for a given sample $x_i$ based on LC is defined as $\text{score}(x_i; \theta_p)_{\text{LC}} = 1 - \max_{\hat{y}} P(\hat{y}|x_i; \theta_p^*)$. In Entropy-Based selection, the entropy of the model's predictions is the focal point. Samples with high entropy indicate situations where $\theta_p^*$ is ambivalent about the correct label. For each sample in $U$, the uncertainty based on entropy is computed as $\text{score}(x_i; \theta_p)_{\text{Entropy}} =$

---

**Algorithm 1** PruneFuse

**Input**: Unlabeled dataset $U$, Initial labeled dataset $s^0$, labeled dataset $L$, original model $\theta$, prune model $\theta_p$, fuse model $\theta_F$, maximum budget $b$, pruning ratio $p$, scored $j^{th}$ data sample $D_j$.
1: Randomly initialize $\theta$
2: $\theta_p \leftarrow \text{Prune}(\theta, p)$ //structure pruning
3: $\theta_p^* \leftarrow \text{Train } \theta_p \text{ on } s^0$
4: $L \leftarrow s^0$

5: **while** $|L| \leq b$ **do**
6:     Compute score$(\mathbf{x}; \theta_p^*)$ for all $x \in U$
7:     $D_k = top_k[D_j \in U]_{j \in [k]}$
8:     Query labels $y_k$ for selected samples $D_k$
9:     Add $(D_k, y_k)$ to $L$
10:     $\theta_p^* \leftarrow \text{Train } \theta_p \text{ on } L$

11: $\theta_F \leftarrow Fuse(\theta, \theta_p^*)$
12: $\theta_F^* \leftarrow \text{Fine-tune } \theta_F \text{ on } L$

13: **return** $L, \theta_F^*$

---

$-\sum_{\hat{y}} P(\hat{y}|x_i; \theta_p^*) \log P(\hat{y}|\mathbf{x}_i; \theta_p^*)$. Subsequently, we select the top-$k$ samples exhibiting the highest uncertainty scores, proposing them as prime candidates for annotation. The objective of Greedy k-centres algorithm is to cherry-pick $k$ centers from the dataset such that the maximum distance of any sample from its nearest center is minimized. The algorithm proceeds in a greedy manner by selecting the first center arbitrarily and then iteratively selecting the next center as the point that is furthest from the current set of centers. The selection is mathematically represented as $x = \arg\max_{x \in U} \min_{c \in \text{centers}} d(x, c)$ where centers is the current set of chosen centers and $d(x, c)$ is the distance between point $x$ and center $c$. While various metrics can be employed to compute this distance, we opt for the Euclidean distance since it is widely used in this context.

## 4.3 Training of Pruned Model

Once we have selected the samples from $U$, they are annotated to obtain their respective labels. These freshly labeled samples are assimilated into the labeled dataset $L$. At the start of each training cycle, a fresh pruned model $\theta_p$ is generated. Training from scratch in every iteration is vital to prevent the model from developing spurious correlations or overfitting to specific samples (Coleman et al., 2019). This fresh start ensures that the model learns genuine patterns in the updated labeled dataset without carrying over potential biases from previous iterations. The training process adheres to a typical deep learning paradigm. Given the dataset $L$ with samples $(x_i, y_i)$, the aim is to minimize the loss function: $\mathcal{L}(\theta_p, L) = \frac{1}{|L|} \sum_{i=1}^{|L|} \mathcal{L}_i(\theta_p, x_i, y_i)$, where $\mathcal{L}_i$ denotes the individual loss for the sample $x_i$. Training unfolds over multiple iterations (or epochs). In each iteration, the weights of $\theta_p$ are updated using backpropagation with an optimization algorithm like stochastic gradient descent (SGD).

This process is inherently iterative as in Active Learning. After each round of training, new samples are chosen, annotated, and the model is reinitialized and retrained from scratch. This cycle persists until certain stopping criteria, e.g. labeling budget or desired performance, are met. With the incorporation of new labeled samples at every stage, $\theta_p^*$ progressively refines its performance, becoming better suited for the subsequent data selection phase.

## 4.4 Fusion with the Original Model

After achieving the predetermined budget, the next phase is to integrate the insights from the trained pruned model $\theta_p^*$ into the untrained original model $\theta$. This step is crucial, as it amalgamates the learned knowledge from the pruned model with the expansive architecture of the original model, aiming to harness the best of both worlds.

**Rationale for Fusion.** Traditional pruning and fine-tuning methods often involve training a large model, pruning it down, and then fine-tuning the smaller model. While this is effective, it does not fully exploit the potential benefits of the larger, untrained model. The primary reason is that the pruning process might discard useful structures and connections within the original model that were not yet leveraged during initial training. By fusing the trained pruned model with the untrained original model, we aim to create a model that combines the learned knowledge by $\theta_p^*$ with the broader, unexplored model $\theta$.

**The Fusion Process.** Fusion is executed by transferring the weights from the trained pruned model's weight matrix $\theta_p^*$ to the corresponding locations within the weight matrix of the untrained original model $\theta$. This results in a new, fused weight matrix:

$$\theta_F = Fuse(\theta, \theta_p^*)$$

Let's represent a model $\theta$ as a sequence of layers, where each layer $L$ consists of filters (for CNNs). We can denote the $i^{th}$ filter of layer $j$ in model $\theta$ as $F_{i,j}^\theta$. Given: $\theta$ is the original untrained model and $\theta_p^*$ is the trained pruned model. For a specific layer $j$, $\theta$ has a set of $n$ filters $\{F_{1,j}^\theta, F_{2,j}^\theta, ...F_{n,j}^\theta\}$ and $\theta_p^*$ has a set of $m$ filters $\{F_{1,j}^{\theta_p^*}, F_{2,j}^{\theta_p^*}, ...F_{m,j}^{\theta_p^*}\}$ where $m \leq n$ due to pruning. The fusion process for layer $j$ can be mathematically represented as:

$$F_{i,j}^{\theta_F} = \begin{cases} F_{i,j}^{\theta_p^*} & \text{if } F_{i,j}^{\theta_p^*} \text{ exists} \\ F_{i,j}^\theta & \text{otherwise} \end{cases}$$

Where $F_{i,j}^{\theta_F}$ is the $i^{th}$ filter of layer $j$ in the fused model $\theta_F$. Another approach is that the pruned weights are dispersed over the whole network (an expansion fusion), however, it requires a more complex mapping function. Assuming we have a dispersion function $D$ that maps the filters of $\theta_p^*$ to multiple filters in $\theta$, the fusion can be represented as:

$$F_{i,j}^{\theta_F} = \begin{cases} D(F_{i,j}^{\theta_p^*}) & \text{if } F_{i,j}^{\theta_p^*} \text{ exists} \\ F_{i,j}^\theta & \text{otherwise} \end{cases}$$

Here, $D$ is the dispersion function that averages weights, distributes them across multiple filters, or uses other strategies to disperse the pruned weights across the original model's architecture.

**Advantages of Retaining Unaltered Weights:** By copying weights from the trained pruned model $\theta_p^*$ into their corresponding locations within the untrained original model $\theta$, and leaving the remaining weights of $\theta$ yet to be trained, we create a unique blend. The weights from $\theta_p^*$ encapsulate the knowledge acquired during training, providing a foundation. Meanwhile, the rest of the untrained weights in $\theta$ still have their initial values, offering an element of randomness. This duality fosters a richer exploration of the loss landscape during subsequent training. Fig. 2 illustrates the transformation in training trajectories resulting from the fusion process. The trained weights of $\theta_p^*$ provides a better initialization, while the unaltered weights serve as gateways to unexplored regions in the loss landscape. This strategic combination in the fused model $\theta_F$ enables the discovery of potentially superior solutions that neither the pruned nor the original model might have discovered on their own.

## 4.5 Refinement via Knowledge Distillation

After the fusion process, our resultant model, $\theta_F$, embodies a synthesis of insights from both the trained pruned model $\theta_p^*$ and the original model $\theta$. Although we show that PruneFuse based on

discussed strategy above outperforms baseline active learning, to further optimize and enhance this amalgamated knowledge, we engage in a fine-tuning phase making use of Knowledge Distillation (KD). KD traditionally facilitates a student model to learn and emulate the behavior of a large complex teacher model. While this technique has been employed in various scenarios, its application in our context is unique and particularly advantageous. Given the seamless integration capability of our pruned model, KD stands as a robust tool to complement the learning process. In essence, it's not merely about transferring knowledge; it's about leveraging the insights from $\theta_p^*$ to enrich the training of fused model $\theta_F$. During the fine-tuning phase, we can make use of two losses. The first is the Cross-Entropy Loss, which quantifies the divergence between the predictions of $\theta_F$ and the actual labels in dataset $L$. The second is the Distillation Loss, which measures the difference in the softened logits of $\theta_F$ and $\theta_p^*$. These softened logits are derived by tempering logits of $\theta_p^*$, which in our case is the teacher model, with a temperature parameter before applying the softmax function. The composite loss for the fine-tuning phase is formulated as a weighted average of the Cross-Entropy and Distillation losses. The iterative enhancement of $\theta_F$ is governed by:

$$\theta_F^{(t+1)} = \theta_F^{(t)} - \alpha \nabla_{\theta_F^{(t)}} \left( \lambda \mathcal{L}_{\text{Cross Entropy}}(\theta_F^{(t)}, L) + (1 - \lambda)\mathcal{L}_{\text{Distillation}}(\theta_F^{(t)}, \theta_p^*) \right)$$

Here $\alpha$ represents the learning rate, while $\lambda$ functions as a coefficient to balance the contributions of the two losses. Incorporating KD in the fine-tuning phase provides a structured approach to harness the insights of the pruned model $\theta_p^*$. By doing so, we aim to ensure that the fused model $\theta_F$ not only retains the trained weights of pruned model but also reinforce this knowledge iteratively, optimizing the performance of $\theta_F$ in subsequent tasks.

### 4.6 PruneFuse V2: Iterative Pruning of Fused Model

PruneFuse V2 introduces a strategy to update pruned model, $\theta_p$, from the trained fused model $\theta_F^*$ at predefined intervals $T_{\text{sync}}$. Algorithm 2 describes the PruneFuse V2 methodology precisely. In each active learning cycle, $\theta_p$, obtained by pruning a randomly initialized network, is trained on the labeled dataset $L$ and subsequently employed to score the unlabeled data $U$. At every $T_{\text{sync}}$ cycle, the pruned model $\theta_p$, is obtained by pruning the trained fused model $\theta_F^*$, which will be fine-tune with labeled dataset $L$ to get $\theta_p^*$ and then employed to score the unlabeled data $U$ in the subsequent rounds.

By periodically synchronizing the pruned model with the fused model at regular $T_{\text{sync}}$ intervals, PruneFuse V2 effectively balances computational efficiency with data selection precision compared to PruneFuse Algorithm 1. This itera-

---

**Algorithm 2** PruneFuse V2: Iterative Fused Pruning for Efficient Data Selection

**Input**: AL rounds $R$, Sync interval $T_{sync}$, $U$, $s^0$, $L$, $\theta$, $\theta_p$, $\theta_F$, $b$, $p$.
1: $\theta_p \leftarrow \text{Prune}(\theta, p)$ // Random pruning
2: $\theta_p^* \leftarrow \text{Train } \theta_p \text{ on } s^0$
3: $L \leftarrow s^0$
4: **for** $r = 1$ to $R$ **do**
5:     Select $D_k$ from $U$ using $\text{score}(x; \theta_p^*)$
6:     Add $(D_k, y_k)$ to $L$ rounds
7:     Train $\theta_p^*$ on $L$
8:     **if** $r \% T_{sync} == 0$ **then**
9:         $\theta_F \leftarrow \text{Fuse}(\theta, \theta_p^*)$ // Fuse after $T_{sync}$
10:         $\theta_F^* \leftarrow \text{Fine-tune } \theta_F \text{ on } L$
11:         $\theta_p \leftarrow \text{Prune}(\theta_F^*, p)$ // Prune fused model
12:         $\theta_p^* \leftarrow \text{Fine-tune } \theta_p \text{ on } L$
13: **return** $L, \theta_F^*$

---

tive refinement process enables the pruned model to leverage the robust architecture of fused model, allowing it to evolve dynamically with each cycle and leading to continuous performance improvements. As a result, PruneFuse V2 achieves a more optimal trade-off between accuracy and efficiency compared to the Algorithm 1, enhancing the active learning process while maintaining computational viability. We provide detailed error analysis of this strategy in Supplementary Materials.

## 5 Experiments

### 5.1 Experimental Setup

**Datasets.** The effectiveness of our approach is assessed on different image classification datasets; CIFAR-10 (Krizhevsky et al., 2009), CIFAR-100 (Krizhevsky et al., 2009), TinyImageNet-200 (Le and Yang, 2015), and ImageNet-1K (Deng et al., 2009) with an input size of $32 \times 32 \times 3$ for CIFAR-10 and CIFAR-100, $64 \times 64 \times 3$ for TinyImageNet, and $224 \times 224 \times 3$ for ImageNet-1K. CIFAR-10 is partitioned into 50,000 training and 10,000 test samples, CIFAR-100 contains 100 classes and has 500 training and 100 testing samples per class, whereas TinyImageNet-200 contains 200 classes with

| Method | Params (Million) | CIFAR-10 Label Budget ($b$) | | | | | CIFAR-100 Label Budget ($b$) | | | | | Params (Million) | Tiny-ImageNet-200 Label Budget ($b$) | | | | | ImageNet-1K Label Budget ($b$) | | | | |
|---|---|---|---|---|---|---|---|---|---|---|---|---|---|---|---|---|---|---|---|---|---|---|
| | | 10% | 20% | 30% | 40% | 50% | 10% | 20% | 30% | 40% | 50% | | 10% | 20% | 30% | 40% | 50% | 10% | 20% | 30% | 40% | 50% |
| Baseline (AL) | 0.85 | 80.53 | 87.74 | 90.85 | 92.24 | 93.00 | 35.99 | 52.99 | 59.29 | 63.68 | 66.72 | 25.56 | 14.86 | 33.62 | 43.96 | 49.86 | 54.65 | 52.97 | 64.52 | 69.30 | 71.98 | 73.56 |
| PruneFuse ($p=0.5$) | 0.21 | 80.92 | 88.35 | 91.44 | 92.77 | 93.65 | 40.26 | 53.90 | 60.80 | 64.98 | 67.87 | 6.10 | 18.71 | 39.70 | 47.41 | 51.84 | 55.89 | 55.03 | 65.12 | 69.72 | 72.07 | 73.86 |
| PruneFuse ($p=0.6$) | 0.13 | 80.58 | 87.79 | 90.94 | 92.58 | 93.08 | 37.82 | 52.65 | 60.08 | 63.7 | 66.89 | 3.92 | 19.25 | 38.84 | 47.02 | 52.09 | 55.29 | 54.69 | 65.13 | 69.74 | 72.48 | 74.00 |
| PruneFuse ($p=0.7$) | 0.07 | 80.19 | 87.88 | 90.70 | 92.44 | 93.40 | 36.76 | 52.15 | 59.33 | 63.65 | 66.84 | 2.23 | 18.32 | 39.24 | 46.45 | 52.02 | 55.63 | 53.73 | 64.43 | 68.95 | 71.81 | 73.84 |
| PruneFuse ($p=0.8$) | 0.03 | 80.11 | 87.58 | 90.50 | 92.42 | 93.32 | 36.49 | 50.98 | 58.53 | 62.87 | 65.85 | 1.02 | 18.34 | 37.86 | 47.15 | 51.77 | 55.18 | 53.08 | 64.00 | 69.00 | 71.79 | 73.64 |

Table 1: **Performance Comparison** of Baseline and PruneFuse on CIFAR-10, CIFAR-100 and Tiny ImageNet-200. This table summarizes the test accuracy of final models (original in case of AL and Fused in PruneFuse) for various pruning ratios ($p$) and labeling budgets($b$). Params corresponds to the number of parameters of the data selector model. Least Confidence is used as a metric for subset selection and different architectures (ResNet-56 for CIFAR-10 and CIFAR-100 while ResNet-50 for Tiny-ImageNet-200 and ImageNet-1K) are utilized.

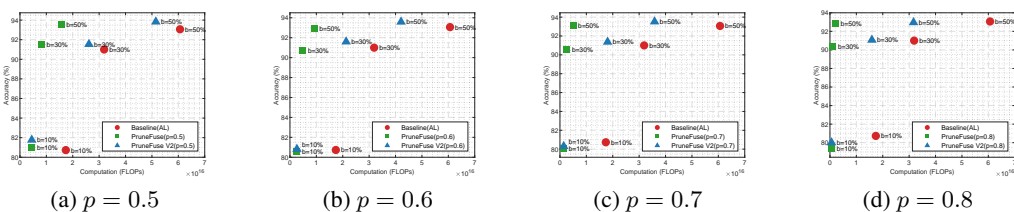

| (a) $p = 0.5$ | (b) $p = 0.6$ | (c) $p = 0.7$ | (d) $p = 0.8$ |
|---|---|---|---|

Figure 3: **Computation Comparison of PruneFuse and Baseline (Active Learning).** This figure illustrates the total number of FLOPs utilized by PruneFuse, compared to the baseline Active Learning method, for selecting subsets with specific labeling budgets $b = 10\%, 30\%, 50\%$. The experiments are conducted on the CIFAR-10 dataset using the ResNet-56 architecture. Subfigures (a), (b), (c), and (d) correspond to different pruning ratios (0.5, 0.6, 0.7, and 0.8, respectively).

500 training, 50 validation, and 50 test samples per class. ImageNet-1K, a more challenging dataset, consists of 1,000 classes with approximately 1.2 million training images and 50,000 validation images, providing a comprehensive benchmark for evaluating large-scale image classification models.

**Implementation Details.** We used ResNet-50, ResNet-56, ResNet-110, and ResNet-164 architecture in our experiments. We pruned these architectures using the Torch-Prunnig library (Fang et al., 2023) for different pruning ratios $p = 0.5, 0.6, 0.7$, and $0.8$ to get the pruned architectures. We trained the model for 181 epochs following the setup in Coleman et al. (2019) for CIFAR-10 and CIFAR-100 and for 100 epochs for TinyImageNet-200 and ImageNet-1K. We use the mini-batch of 128 for CIFAR-10 and CIFAR-100 and 256 for TinyImageNet-200 and ImageNet-1K. For all the experiments SGD is used as an optimizer (further details are provided in Suplementary Materials A.3). We took Active Learning (AL) as a baseline for the proposed technique and initially, we started by randomly selecting 2% of the data. For the first round, we added 8% from the unlabeled set, then 10% in each subsequent round, until reaching the label budget, $b$. After each round, we retrained the models from scratch, as described in the methodology. All experiments are carried out independently 3 times and then the average is reported.

## 5.2 RESULTS AND DISCUSSIONS

**Main Experiments.** We compare the performance of the PruneFuse with the baseline AL across different model architectures, datasets, labeling budgets, and data selection metrics (detailed results are provided in Supplementary Materials A.4). These experiments aim to demonstrate superior generalization performance and computational efficiency. Table 1 summarizes the performance of baseline and different variants of PruneFuse on various datasets. Results show that PruneFuse consistently outperforms the baseline in most cases. The accuracy advantage in case of high pruning ratio, e.g. in the case of $p = 0.7$, demonstrates the effectiveness of superior data selection performance and fusion. Fig. 3 (a), (b), (c), and (d) shows the trade-off between accuracy and the computational complexity of the baseline and PruneFuse variants in terms of Floating Point Operations (FLOPs) for different labeling budgets. The FLOPs are computed for the whole training duration of the pruned network and the selection process for a given budget. Different variants of PruneFuse, with pruning ratios $p = 0.5$, $p = 0.6$, $p = 0.7$, and $p = 0.8$, offer users the flexibility to choose a version based on their computational resources. For instance, PruneFuse ($p = 0.8$) requires significantly

| Method | Params (Million) | Label Budget ($b$), $T_{sync} = 1$ | | | | | Label Budget ($b$), $T_{sync} = 2$ | | | | |
|---|---|---|---|---|---|---|---|---|---|---|---|
| | | 20% | 30% | 40% | 50% | 60% | 20% | 30% | 40% | 50% | 60% |
| Baseline | 0.85 | 88.51 | 91.46 | 93.04 | 93.61 | 93.83 | 88.51 | 91.46 | 93.04 | 93.61 | 93.83 |
| PruneFuse V2 ($p = 0.5$) | 0.21 | **88.52** | **91.76** | **93.15** | 93.78 | 93.90 | **88.59** | 91.47 | **93.05** | **93.84** | **93.88** |
| PruneFuse V2 ($p = 0.6$) | 0.13 | **88.53** | 91.71 | 93.08 | 93.67 | 93.90 | 88.14 | 91.47 | 92.87 | 93.57 | 93.79 |
| PruneFuse V2 ($p = 0.7$) | 0.07 | 88.37 | **91.47** | 93.00 | 93.33 | 93.69 | 88.41 | 91.51 | 92.67 | 93.46 | 93.72 |

Table 2: Performance of PruneFuse V2 with $T_{sync} = 1$ and $T_{sync} = 2$ for different pruning ratios and label budgets.

| Method | Label Budget ($b$) | | | | |
|---|---|---|---|---|---|
| | 10% | 20% | 30% | 40% | 50% |
| **Baseline (AL)** | 80.53 | 87.74 | 90.85 | 92.94 | 93.00 |
| **BALD** | 80.61 | 88.11 | 91.21 | 92.98 | 93.36 |
| **SVP** | 80.76 | 87.31 | 90.77 | 92.59 | 92.95 |
| **PruneFuse** | 80.92 | 88.35 | 91.44 | 92.77 | 93.65 |
| **PruneFuse V2** | 81.23 | 88.52 | 91.76 | 93.15 | 93.78 |
| **PruneFuse V2 + BALD** | 80.71 | 88.38 | 91.44 | 93.16 | 93.58 |

Table 3: Comparison with Baselines for Resnet-56 on Cifar-10.

| Method | Selection Size (k) | Label Budget ($b$) | | |
|---|---|---|---|---|
| | | 20% | 40% | 60% |
| **Baseline** ($AL$) | 5,000 | 88.51 | 93.04 | 93.83 |
| **PruneFuse V2** | 5,000 | **88.82** | **93.15** | **93.90** |
| **Baseline** ($AL$) | 10,000 | 86.92 | 92.51 | 93.81 |
| **PruneFuse V2** | 10,000 | **87.49** | **93.11** | **94.04** |

Table 4: Ablation study of $k$ on Cifar-10 using ResNet-56 with ($p = 0.5$).

lower computational resources while still achieving good accuracy performance. PruneFuse V2 ($p = 0.5$) strikes an effective balance between accuracy and computation. It consistently provides high accuracy with moderate FLOPs, making it an ideal choice for scenarios where both performance and computational efficiency are critical. Compared to the baseline AL, both PruneFuse and PruneFuse V2 demonstrates superior performance at every label budget, all while reducing the computational cost. Detailed Complexity Analysis and Error Analysis for PruneFuse are provided in Supplementary Materials A.1 and A.2, respectively.

**PruneFuse V2.** We further evaluate PruneFuse V2 and compared it's efficacy against baseline AL. We conducted experiments by varying the synchronization interval $T_{sync}$ to evaluate the impact of the frequency at which the pruned model is fused with the original model. Specifically, we used $T_{sync} = 1$, where the pruned model is updated from the trained fused model after every round, and $T_{sync} = 2$, where this update happens after every two rounds. For a fair comparison, we modified the baseline to continue retraining the network from the previous round, rather than reinitializing it. While this provided a slight improvement for the baseline, PruneFuse still outperformed it by a significant margin.

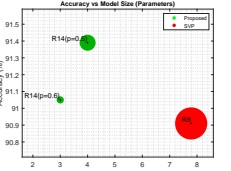 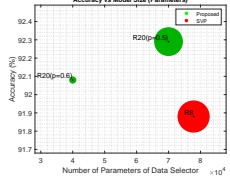

(a) Target Model = ResNet-14     (b) Target Model = ResNet-20

Figure 4: **Comparison of PruneFuse with SVP.** Scatter plot shows final accuracy on target model against the model size for different ResNet models on CIFAR-10, $b = 50\%$. (a) shows ResNet-14 (with $p = 0.5$ and $p = 0.6$) and ResNet-8 models are used as data selectors for PruneFuse and SVP, respectively. While in (b), PruneFuse utilizes ResNet20 (i.e. $p = 0.5$ and $p = 0.6$) and SVP utilizes ResNet-8 models.

As shown in Table 2, $T_{sync} = 1$ leads to better performance due to more frequent updates and refinements of the pruned model. However, $T_{sync} = 2$ also shows strong results with fewer updates, offering a balance between computational efficiency and accuracy. At higher label budgets (e.g., 60%), both approaches perform similarly, indicating that PruneFuse can adapt to different synchronization intervals without significant performance degradation.

These results highlight that while more frequent updates $T_{sync} = 1$ results in better data selection, $T_{sync} = 2$ offers a more computationally efficient alternative without compromising much on accuracy. This flexibility makes PruneFuse an effective solution for a variety of resource-constrained scenarios.

**Comparison with Baselines.** Table 3 delineates a performance comparison of PruneFuse with baseline techniques, including SVP and BALD, across various labeling budgets $b$ for efficient training of a target model (ResNet-56) on the CIFAR-10 dataset. Here, SVP employs a ResNet-20 as its data selector, with a model size of 0.26 M. In contrast, PruneFuse uses a 50% pruned ResNet-56, reducing its data selector size to 0.21 M. BALD similar to baseline AL, uses ResNet-56 for data selection based on Bayesian uncertainty. Performance metrics demonstrate that PruneFuse consistently outperforms

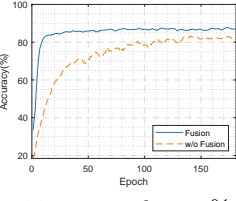 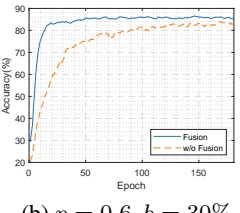 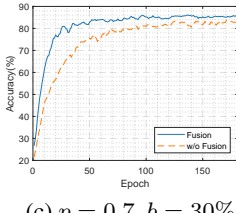

(a) $p = 0.5$, $b = 30\%$      (b) $p = 0.6$, $b = 30\%$      (c) $p = 0.7$, $b = 30\%$

Figure 5: **Impact of Model Fusion on PruneFuse Performance:** This figure compares the accuracy over epochs between fused and non-fused training approaches within the PruneFuse framework. Experiments are conducted using the ResNet-56 on the CIFAR-10. Subfigures (a), (b) and (c) correspond to pruning ratios $p = 0.5$, 0.6 and 0.7, respectively.

SVP across label budgets ranging from 10% to 50%. For example, PruneFuse achieves 80.92% accuracy at a 10% label budget and peaks at 93.65% at 50%, compared to SVP's 80.76% and 92.95%, respectively. Fig. 4 further illustrates the comparison in terms of model sizes. The enhanced PruneFuse V2 shows even greater performance, particularly with $T_{sync} = 1$, where more frequent updates enable it to reach 93.78% accuracy at 50%. This highlights the efficiency of PruneFuse's data selection and fusion process over traditional methods like SVP. BALD, while demonstrating competitive results at higher label budgets (e.g., 93.36% at 50%), remains slightly behind PruneFuse's performance. Nevertheless, BALD can be seamlessly integrated with PruneFuse. This integration, seen in PruneFuse V2 + BALD, capitalizes on the strengths of both methods, yielding improved performance. Notably, PruneFuse V2 + BALD achieves 93.16% accuracy at a 40% label budget, illustrating the potential of combining these approaches for even better results in high-budget scenarios.

**Additional Experiments and Ablation Studies.**

Fig. 5 demonstrates the effect of fusion across various pruning ratios, the models trained with fusion in-place perform better than those trained without fusion, achieving higher accuracy levels at an accelerated pace. The rapid convergence is most notable in initial training phases, where fusion model benefits from the initialization provided by the integration of weights from a trained pruned model $\theta_p^*$ with an untrained model $\theta$. The strategic retention of untrained weights introduces a beneficial stochastic component to the training process, enhancing the model's ability to explore new regions of the parameter space. This dual capability of exploiting prior knowledge and exploring new configurations enables the proposed technique to consistently outperform, making it particularly beneficial in scenarios with sparse label data. Table 4 demonstrates the effect of selection size $k$. PruneFuse V2 consistently outperforms the Baseline AL in terms of selection size indicating the efficacy of the data selection. The impact of different selection metrics (Least Confidence, Entropy, Random, and Greedy K Centers) is presented in Table 5 across both the Baseline and PruneFuse methods. In both cases, the Least Confidence metric surfaces as particularly effective in optimizing label utilization and model performance. The results show that regardless of the label budget and strategy utilized for data selection, PruneFuse consistently performs superior as compared to Baseline. Ablation study of Knoweledge distillation is provides in Suplementary Materials A.6.

| Method | Selection Metric | Label Budget ($b$) | | | | |
|---|---|---|---|---|---|---|
| | | 10% | 20% | 30% | 40% | 50% |
| Baseline ($AL$) | Least Conf | 38.41 | 51.39 | 65.53 | 70.07 | 73.05 |
| | Entropy | 36.65 | 57.58 | 64.98 | 69.99 | 72.90 |
| | Random | 39.31 | 57.53 | 63.84 | 67.75 | 70.79 |
| | Greedy k | 39.76 | 57.40 | 65.20 | 69.25 | 72.91 |
| PruneFuse ($p = 0.5$) | Least Conf | **42.88** | **59.31** | **66.95** | **71.45** | **74.32** |
| | Entropy | **42.99** | **59.32** | **66.83** | **71.18** | **74.43** |
| | Random | **43.72** | **58.58** | **64.93** | **68.75** | **71.63** |
| | Greedy k | **43.61** | **58.38** | **66.04** | **69.83** | **73.10** |

Table 5: Effect of Different Data Selection Metrics on CIFAR-100 using ResNet-164 architecture.

## 6 CONCLUSION

In this work, we present PruneFuse, a novel strategy that integrates pruning with network fusion to optimize the data selection pipeline for deep learning. PruneFuse leverages a small pruned model for data selection, which then seamlessly fuses with the original model, providing fast and better generalization while significantly reducing computational costs. Our extensive evaluations across CIFAR-10, CIFAR-100, Tiny-ImageNet-200, and ImageNet-1K demonstrate that PruneFuse consistently outperforms existing baselines, establishing its efficiency and efficacy. PruneFuse offers a scalable, practical, and flexible solution to enhance the training efficiency of neural networks, particularly in resource-constrained settings.

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

## A  SUPPLEMENTARY MATERIALS

This supplementary material provides additional details, analyses, and results to complement the main paper. The content is organized into the following subsections:

1. **Complexity Analysis** (A.1): A detailed breakdown of the computational complexity of PruneFuse and its components.

2. **Error Analysis for PruneFuse** (A.2): An error analysis outlining theoretical guarantees for the proposed framework.

3. **Implementation Details** (A.3): Specific details about the experimental setup, hyperparameters, and configurations used in our experiments.

4. **Performance Comparison with Different Datasets, Selection Metrics, and Architectures** (A.4): Results demonstrating PruneFuse's adaptability across datasets and architectures.

5. **Ablation Study of Fusion** (A.5): Analysis of the impact of the fusion process on PruneFuse's performance.

6. **Ablation Study of Knowledge Distillation in PruneFuse** (A.6): An evaluation of the role of knowledge distillation in improving performance.

7. **Comparison with SVP** (A.7): A comparison highlighting differences and improvements over the SVP baseline.

8. **Ablation Study on the Number of Selected Data Points ($k$)** (A.8): Investigation of how varying $k$ affects PruneFuse's performance.

9. **Impact of Early Stopping on Performance** (A.9): Evaluation of the utility of early stopping when integrated with PruneFuse.

10. **Performance Comparison Across Architectures and Datasets** (A.10): Additional results comparing PruneFuse's performance on various architectures and datasets.

11. **Performance at Lower Pruning Rates** (A.11): Results demonstrating PruneFuse's effectiveness at lower pruning rates.

12. **Comparison with Recent Coreset Selection Techniques** (A.12): Evaluation of PruneFuse's performance with recent coreset selection methods.

13. **Effect of Various Pruning Strategies and Criterion** (A.13): Analysis of different pruning techniques and criteria on PruneFuse's performance.

14. **Detailed Runtime Analysis of PruneFuse** (A.14): A detailed runtime analysis of PruneFuse compared to baseline methods.

Each section provides additional insights, evaluations, and experiments to further validate and explain the effectiveness of the proposed approach.

## A.1 Complexity Analysis

Given $P$ and $N$ represent the total number of parameters in the pruned and dense model, where $P \ll N$, the computational costs can be summarized as follows:

**Initial Training on $s_0$:**

$$
\begin{aligned}
\text{PruneFuse:} &\quad O\left(|s_0| \times P \times T\right) + O\left(P \times \log P\right) \text{ one time pruning cost} \\
\text{Baseline AL:} &\quad O\left(|s_0| \times N \times T\right)
\end{aligned}
$$

**Data selection round with current labeled pool L:**

$$
\begin{aligned}
\text{PruneFuse:} &\quad O\left(|L| \times P \times T\right) + O\left(|U| \times P\right) \text{ selection} \\
\text{Baseline AL:} &\quad O\left(|L| \times N \times T\right) + O\left(|U| \times N\right) \text{ selection}
\end{aligned}
$$

**Training of the final model on the final labeled set L:**

$$
\begin{aligned}
\text{PruneFuse:} &\quad O\left(|L| \times N \times T\right) + O\left(P\right) \text{ one time fusion cost} \\
\text{Baseline AL:} &\quad O\left(|L| \times N \times T\right)
\end{aligned}
$$

**Total training complexity:**

$$
\begin{aligned}
\text{PruneFuse:} &\quad O\left(|s_0| \times P \times T\right) + O\left(P \times \log P\right) + R \times \left[O\left(|L| \times P \times T\right) + O\left(|U| \times P\right)\right] \\
&\quad + O\left(|L| \times N \times T\right) + O(P) \\
\text{PruneFuse V2:} &\quad O\left(|s_0| \times P \times T\right) + O\left(P \times \log P\right) + R \times \left[O\left(|L| \times P \times T\right) + O\left(|U| \times P\right)\right] \\
&\quad + F_{sync} * \left[O\left(|L| \times N \times T\right) + O(P) + O\left(|L| \times P \times T\right) + O\left(P \times \log P\right)\right] \\
&\quad + O\left(|L| \times N \times T\right) + O(P) \\
\text{Baseline AL:} &\quad O\left(|s_0| \times N \times T\right) + R \times \left[O\left(|L| \times N \times T\right) + O\left(|U| \times N\right)\right] + O\left(|L| \times N \times T\right)
\end{aligned}
$$

Here $T$ represents the total number of Epochs for a training round of AL which in our case is set to 181. $U$ is the whole unlabeled dataset and $R$ represents the total number of AL rounds. $F_{sync}$ represent the frequency of iterative pruning based on the fused model.

We can see that the major training costs in Active Learning (AL) arise from the repeated use of a large, dense model, which significantly increases computational expenses, especially across multiple rounds of data selection. By using a smaller surrogate (pruned model) for these rounds, as implemented in PruneFuse, the training cost and overall computation are reduced substantially. This approach leads to a more efficient and cost-effective data selection process, allowing for better resource utilization while maintaining high performance.

## A.2 Error Analysis for PruneFuse

We analyze the error in PruneFuse by decomposing it into two components: *selection error*, arising from training the pruned model on a subset $s_p$ of the full dataset $D$, and *pruning error*, resulting from the reduced capacity of the pruned model $\theta_p$. We demonstrate how the synchronization frequency $F_{\text{sync}}$ controls both errors and present a convergence result under reasonable assumptions.

The optimization problem is formulated as:

$$
\min_{s_p} \left| \mathbb{E}_{(x,y) \in s_p} \left[l(x, y; \theta_p)\right] - \mathbb{E}_{(x,y) \in D} \left[l(x, y; \theta)\right] \right| \tag{3}
$$

where $s_p \subset D$ is the selected subset, $\theta_p$ is the pruned model, and $\theta$ is the full model. Our goal is to minimize the difference in expected loss between the pruned model on the subset $s_p$ and the full model on the full dataset $D$.

We make the following assumptions to formalize the error bounds:

**Assumption 1.** *The loss function $l(x, y; \theta)$ is Lipschitz continuous with respect to the model parameters $\theta$, with constant $L$:*

$$|l(x, y; \theta_1) - l(x, y; \theta_2)| \le L\|\theta_1 - \theta_2\|$$

**Assumption 2.** *The pruned subset $s_p$ is assumed to be an i.i.d. sample from the full dataset $D$, and the expected loss over $s_p$ approximates that over $D$ with high probability. Specifically, there exists a constant $\delta$ such that:*

$$\left|\mathbb{E}_{(x,y) \in s_p}[l(x, y; \theta)] - \mathbb{E}_{(x,y) \in D}[l(x, y; \theta)]\right| \le \delta$$

**Assumption 3.** *After each synchronization step, the pruned model $\theta_p$ is updated to reduce its distance from the full model $\theta$. Specifically, the synchronization reduces the distance by a factor $\alpha$, where $0 < \alpha < 1$, meaning:*

$$\|\theta_p^{t+1} - \theta\| \le \alpha\|\theta_p^t - \theta\|$$

**Selection Error.** The *selection error*, denoted $\mathcal{E}_{\text{sel}}$, arises from training the pruned model on the subset $s_p$ rather than the full dataset $D$. Using assumptions 1 and 2, we can bound this error as:

$$\mathcal{E}_{\text{sel}} \le L\|\theta_p - \theta\| + \delta \tag{4}$$

where $\delta$ is the subset approximation error and $L$ is the Lipschitz constant of the loss function.

Furthermore, since $\theta_p$'s representational power improves with synchronization, we express $\|\theta_p - \theta\|$ as decreasing over time due to synchronization. The representational power of $\theta_p$ improves with synchronization, so:

$$\mathcal{E}_{\text{sel}} \le \frac{C_0}{F_{\text{sync}}}L + \delta \tag{5}$$

Where $C_0$ represents the initial distance between the pruned model $\theta_p$ and the full model $\theta$ and $F_{sync}$ is the frequency of synchronization.

**Pruning Error.** The *pruning error*, denoted $\mathcal{E}_{\text{prune}}$, arises from the reduced capacity of the pruned model $\theta_p$. By Assumption 1 and 3, the pruning error can be controlled by the distance between $\theta_p$ and $\theta$. The error is reduced after each synchronization step as:

$$\mathcal{E}_{\text{prune}} \le \frac{C_\theta}{F_{\text{sync}}} \tag{6}$$

where $C_\theta$ is a constant reflecting the magnitude of the pruning error, and $F_{\text{sync}}$ is the synchronization frequency. More frequent synchronization decreases the pruning error.

**Total Error.** The total error $\mathcal{E}_{\text{total}}$ is the sum of the selection error $\mathcal{E}_{\text{sel}}$ and the pruning error $\mathcal{E}_{\text{prune}}$. Substituting the bounds for each component, we obtain:

$$\mathcal{E}_{\text{total}} = \mathcal{E}_{\text{sel}} + \mathcal{E}_{\text{prune}} \le \frac{C_0 L + C_\theta}{F_{\text{sync}}} + \delta \tag{7}$$

Furthermore, under assumption A3, synchronization leads to the following convergence result for the distance between $\theta_p$ and $\theta$:

$$\|\theta_p^t - \theta\| \le \alpha^t\|\theta_p^0 - \theta\| \tag{8}$$

where $t$ is the number of synchronization steps. Thus, after $t$ steps, the pruned model converges to the full model with an exponential rate controlled by $\alpha$.

The total error decreases as the synchronization frequency $F_{\text{sync}}$ increases. Moreover, under reasonable assumptions, the pruned model $\theta_p$ converges to the full model $\theta$ over time with an exponential rate. The bound shows that synchronization not only reduces the pruning error but also improves the pruned model's ability to generalize on the selected subset, minimizing the selection error.

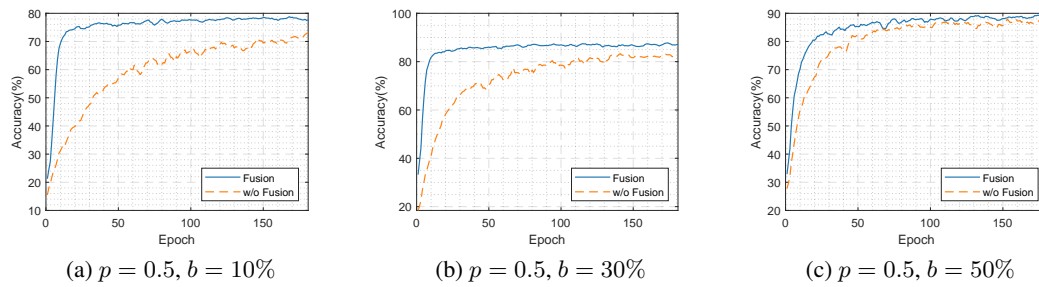

(a) $p = 0.5$, $b = 10\%$     (b) $p = 0.5$, $b = 30\%$     (c) $p = 0.5$, $b = 50\%$

Figure 6: **Ablation Study of Fusion on PruneFuse** ($p = 0.5$). Experiments are performed on ResNet-56 architecture with CIFAR-10.

### A.3    IMPLEMENTATION DETAILS.

We used ResNet-50, ResNet-56, ResNet-110, and ResNet-164 architecture in our experiments. We pruned these architectures using the Torch-Prunnig library (Fang et al., 2023) for different pruning ratios $p = 0.5, 0.6, 0.7,$ and $0.8$ to get the pruned architectures. For CIFAR-10 and CIFAR-100, the models were trained for 181 epochs, with an epoch schedule of [1, 90, 45, 45], and corresponding learning rates of [0.01, 0.1, 0.01, 0.001], using a momentum of 0.9 and weight decay of 0.0005. For TinyImageNet-200 and ImageNet-1K, the models were trained over an epoch schedule of [1, 1, 1, 1, 1, 25, 30, 20, 20], with learning rates of [0.0167, 0.0333, 0.05, 0.0667, 0.0833, 0.1, 0.01, 0.001, 0.0001], a momentum of 0.9, and weight decay of 0.0001. We use the mini-batch of 128 for CIFAR-10 and CIFAR-100 and 256 for TinyImageNet-200 and ImageNet-1K. For all the experiments SGD is used as an optimizer. We set the knowledge distillation coefficient $\lambda$ to 0.3. We took Active Learning (AL) as a baseline for the proposed technique and initially, we started by randomly selecting 2% of the data. For the first round, we added 8% from the unlabeled set, then 10% in each subsequent round, until reaching the label budget, $b$. After each round, we retrained the models from scratch, as described in the methodology. All experiments are carried out independently 3 times and then the average is reported.

### A.4    PERFORMANCE COMPARISON WITH DIFFERENT DATASETS, SELECTION METRICS, AND ARCHITECTURES

To comprehensively evaluate the effectiveness of PruneFuse, we conducted additional experiments comparing its performance with baseline utilizing other data selection metrics such as Least Confidence, Entropy, and Greedy k-centers. Results are shown in Tables 6, 7, and 8 for various architectures and labeling budgets. In all cases, our results demonstrate that PruneFuse mostly outperforms the baseline using these traditional metrics across various datasets and model architectures, highlighting the robustness of PruneFuse in selecting the most informative samples efficiently.

### A.5    ABLATION STUDY OF FUSION

The fusion process is a critical component of the PruneFuse methodology, designed to integrate the knowledge gained by the pruned model into the original network. Our experiments reveal that models trained with the fusion process exhibit significantly better performance and faster convergence compared to those trained without fusion. By initializing the original model with the weights from the trained pruned model, the fused model benefits from an optimized starting point, which enhances its learning efficiency and generalization capability. Fig. 6, 7 and 8 illustrates the training trajectories and accuracy improvements when fusion takes places, demonstrating the tangible benefits of this initialization. These results underscore the importance of the fusion step in maximizing the overall performance of the PruneFuse framework.

### A.6    ABLATION STUDY OF KNOWLEDGE DISTILLATION IN PRUNEFUSE

Table 10 demonstrates the effect of Knowledge Distillation on the PruneFuse technique relative to the baseline Active Learning (AL) method across various experimental configurations and label budgets

| Method | Selection Metric | Label Budget ($b$) | | | | |
|---|---|---|---|---|---|---|
| | | 10% | 20% | 30% | 40% | 50% |
| Baseline AL | Least Conf | 80.53 ± 0.20 | 87.74 ± 0.15 | 90.85 ± 0.11 | 92.24 ± 0.16 | 93.00 ± 0.11 |
| | Entropy | 80.14 ± 0.41 | 87.63 ± 0.10 | 90.80 ± 0.36 | 92.51 ± 0.34 | 92.98 ± 0.03 |
| | Random | 78.55 ± 0.38 | 85.26 ± 0.21 | 88.13 ± 0.35 | 89.81 ± 0.15 | 91.20 ± 0.05 |
| | Greedy k | 79.63 ± 0.83 | 86.46 ± 0.27 | 90.09 ± 0.20 | 91.9 ± 0.08 | 92.80 ± 0.08 |
| PruneFuse $p = 0.5$ | Least Conf | 80.92 ± 0.41 | 88.35 ± 0.33 | 91.44 ± 0.15 | 92.77 ± 0.03 | 93.65 ± 0.14 |
| | Entropy | 81.08 ± 0.16 | 88.74 ± 0.10 | 91.33 ± 0.04 | 92.78 ± 0.04 | 93.48 ± 0.04 |
| | Random | 80.43 ± 0.27 | 86.28 ± 0.37 | 88.75 ± 0.17 | 90.36 ± 0.02 | 91.42 ± 0.12 |
| | Greedy k | 79.85 ± 0.68 | 86.96 ± 0.38 | 90.20 ± 0.16 | 91.82 ± 0.14 | 92.89 ± 0.14 |
| PruneFuse $p = 0.6$ | Least Conf | 80.58 ± 0.33 | 87.79 ± 0.20 | 90.94 ± 0.13 | 92.58 ± 0.31 | 93.08 ± 0.42 |
| | Entropy | 80.96 ± 0.16 | 87.89 ± 0.45 | 91.22 ± 0.28 | 92.56 ± 0.19 | 93.19 ± 0.26 |
| | Random | 79.19 ± 0.57 | 85.65 ± 0.29 | 88.27 ± 0.18 | 90.13 ± 0.24 | 91.01 ± 0.28 |
| | Greedy k | 79.54 ± 0.48 | 86.16 ± 0.60 | 89.50 ± 0.29 | 91.35 ± 0.06 | 92.39 ± 0.22 |
| PruneFuse $p = 0.7$ | Least Conf | 80.19 ± 0.45 | 87.88 ± 0.05 | 90.70 ± 0.21 | 92.44 ± 0.24 | 93.40 ± 0.11 |
| | Entropy | 79.73 ± 0.87 | 87.85 ± 0.25 | 90.94 ± 0.29 | 92.41 ± 0.23 | 93.39 ± 0.20 |
| | Random | 78.76 ± 0.23 | 85.50 ± 0.11 | 88.31 ± 0.19 | 89.94 ± 0.24 | 90.87 ± 0.17 |
| | Greedy k | 78.93 ± 0.15 | 85.85 ± 0.41 | 88.96 ± 0.07 | 90.93 ± 0.19 | 92.23 ± 0.08 |
| PruneFuse $p = 0.8$ | Least Conf | 80.11 ± 0.28 | 87.58 ± 0.14 | 90.50 ± 0.08 | 92.42 ± 0.41 | 93.32 ± 0.14 |
| | Entropy | 79.83 ± 1.13 | 87.50 ± 0.54 | 90.52 ± 0.24 | 92.24 ± 0.13 | 93.15 ± 0.10 |
| | Random | 78.77 ± 0.66 | 85.64 ± 0.13 | 88.45 ± 0.33 | 89.88 ± 0.14 | 91.21 ± 0.43 |
| | Greedy k | 78.23 ± 0.37 | 85.59 ± 0.25 | 88.60 ± 0.19 | 90.11 ± 0.11 | 91.31 ± 0.08 |

(a) CIFAR-10 using ResNet-56 architecture.

| Method | Selection Metric | Label Budget ($b$) | | | | |
|---|---|---|---|---|---|---|
| | | 10% | 20% | 30% | 40% | 50% |
| Baseline AL | Least Conf | 35.99 ± 0.80 | 52.99 ± 0.56 | 59.29 ± 0.46 | 63.68 ± 0.53 | 66.72 ± 0.33 |
| | Entropy | 37.57 ± 0.51 | 52.64 ± 0.76 | 58.87 ± 0.38 | 63.97 ± 0.17 | 66.78 ± 0.27 |
| | Random | 37.06 ± 0.64 | 51.62 ± 0.21 | 58.77 ± 0.65 | 62.05 ± 0.02 | 64.63 ± 0.16 |
| | Greedy k | 38.28 ± 1.11 | 52.43 ± 0.24 | 58.96 ± 0.16 | 63.56 ± 0.30 | 66.30 ± 0.31 |
| PruneFuse $p = 0.5$ | Least Conf | 40.26 ± 0.95 | 53.90 ± 1.06 | 60.80 ± 0.44 | 64.98 ± 0.4 | 67.87 ± 0.17 |
| | Entropy | 38.59 ± 1.67 | 54.01 ± 1.17 | 60.52 ± 0.19 | 64.83 ± 0.27 | 67.67 ± 0.33 |
| | Random | 39.43 ± 0.99 | 54.60 ± 0.64 | 60.13 ± 0.96 | 63.91 ± 0.39 | 66.02 ± 0.3 |
| | Greedy k | 39.83 ± 2.44 | 54.35 ± 0.41 | 60.40 ± 0.23 | 64.22 ± 0.25 | 66.89 ± 0.16 |
| PruneFuse $p = 0.6$ | Least Conf | 37.82 ± 0.83 | 52.65 ± 0.4 | 60.08 ± 0.22 | 63.7 ± 0.25 | 66.89 ± 0.46 |
| | Entropy | 38.01 ± 0.79 | 51.91 ± 0.56 | 59.18 ± 0.31 | 63.53 ± 0.25 | 66.88 ± 0.18 |
| | Random | 38.27 ± 0.81 | 52.85 ± 1.22 | 58.68 ± 0.68 | 62.28 ± 0.22 | 65.2 ± 0.48 |
| | Greedy k | 38.44 ± 0.98 | 52.85 ± 0.74 | 59.36 ± 0.57 | 63.36 ± 0.75 | 66.12 ± 0.38 |
| PruneFuse $p = 0.7$ | Least Conf | 36.76 ± 0.63 | 52.15 ± 0.53 | 59.33 ± 0.17 | 63.65 ± 0.36 | 66.84 ± 0.43 |
| | Entropy | 36.95 ± 1.03 | 50.64 ± 0.33 | 58.45 ± 0.36 | 62.27 ± 0.27 | 65.88 ± 0.28 |
| | Random | 37.30 ± 1.24 | 51.66 ± 0.21 | 58.79 ± 0.13 | 62.67 ± 0.29 | 65.08 ± 0.08 |
| | Greedy k | 38.88 ± 2.18 | 52.02 ± 0.77 | 58.66 ± 0.19 | 61.39 ± 0.11 | 65.28 ± 0.65 |
| PruneFuse $p = 0.8$ | Least Conf | 36.49 ± 0.20 | 50.98 ± 0.54 | 58.53 ± 0.50 | 62.87 ± 0.13 | 65.85 ± 0.32 |
| | Entropy | 36.02 ± 1.30 | 51.23 ± 0.23 | 57.44 ± 0.11 | 62.65 ± 0.46 | 65.76 ± 0.30 |
| | Random | 37.37 ± 0.85 | 52.06 ± 0.47 | 58.19 ± 0.30 | 62.19 ± 0.45 | 64.77 ± 0.29 |
| | Greedy k | 37.04 ± 0.09 | 49.84 ± 0.49 | 56.13 ± 0.20 | 60.24 ± 0.42 | 62.92 ± 0.44 |

(b) CIFAR-100 using ResNet-56 architecture.

Table 6: **Performance Comparison** of Baseline and PruneFuse on CIFAR-10 and CIFAR-100 with ResNet-56 architecture. This table summarizes the test accuracy of final models (original in case of AL and Fused in PruneFuse) for various pruning ratios ($p$), labeling budgets ($b$), and data selection metrics.

on CIFAR-10 and CIFAR-100 datasets, using different ResNet architectures. The results indicate that PruneFuse consistently outperforms the baseline method, both with and without incorporating Knowledge Distillation (KD) from a trained pruned model. This superior performance is attributed to the innovative fusion strategy inherent to PruneFuse, where the original model is initialized using weights from a previously trained pruned model. The proposed approach gives the fused model an optimized starting point, enhancing its ability to learn more efficiently and generalize better. The impact of this strategy is evident across different label budgets and architectures, demonstrating its effectiveness and robustness.

| Method | Selection Metric | Label Budget ($b$) | | | | |
|---|---|---|---|---|---|---|
| | | 10% | 20% | 30% | 40% | 50% |
| Baseline $AL$ | Least Conf. | 80.74 ± 0.04 | 87.80 ± 0.09 | 91.50 ± 0.09 | 93.19 ± 0.14 | 93.68 ± 0.17 |
| | Entropy | 79.81 ± 0.18 | 88.46 ± 0.30 | 91.30 ± 0.15 | 92.83 ±0.30 | 93.47 ± 0.31 |
| | Random | 79.99 ± 0.10 | 85.63 ± 0.03 | 88.07 ± 0.31 | 90.40 ± 0.42 | 91.42 ± 0.26 |
| | Greedy k | 78.69 ± 0.58 | 87.46 ±0.20 | 90.72 ± 0.14 | 92.55 ±0.14 | 93.44 ± 0.07 |
| PruneFuse $p = 0.5$ | Least Conf. | 81.24 ± 0.43 | 88.70 ± 0.15 | 92.02 ± 0.10 | 93.32 ± 0.13 | 94.07 ± 0.06 |
| | Entropy | 81.45 ± 0.39 | 88.90 ± 0.11 | 92.13 ± 0.15 | 93.49 ± 0.16 | 94.07 ± 0.05 |
| | Random | 80.08 ± 0.86 | 86.52 ± 0.14 | 89.48 ± 0.16 | 90.82 ± 0.21 | 91.79 ± 0.04 |
| | Greedy k | 80.40 ± 0.09 | 87.77 ± 0.13 | 90.74 ± 0.09 | 92.48 ± 0.22 | 93.53 ± 0.22 |
| PruneFuse $p = 0.6$ | Least Conf. | 81.12 ± 0.34 | 88.33 ± 0.31 | 91.57 ± 0.03 | 93.25 ± 0.21 | 93.90 ± 0.17 |
| | Entropy | 80.02 ± 0.41 | 88.49 ± 0.18 | 91.51 ± 0.14 | 93.03 ± 0.11 | 93.94 ± 0.12 |
| | Random | 78.55 ± 0.42 | 85.94 ± 0.34 | 88.77 ± 0.10 | 90.66 ± 0.20 | 92.02 ± 0.03 |
| | Greedy k | 79.44 ± 0.28 | 87.05 ± 0.63 | 90.30 ± 0.15 | 92.15 ± 0.12 | 93.22 ± 0.04 |
| PruneFuse $p = 0.7$ | Least Conf. | 79.93 ± 0.06 | 88.04 ± 0.23 | 91.51 ± 0.34 | 92.90 ± 0.02 | 93.82 ± 0.09 |
| | Entropy | 80.16 ± 0.27 | 87.78 ± 0.52 | 91.21 ± 0.13 | 92.99 ± 0.13 | 93.81 ± 0.12 |
| | Random | 79.41 ± 0.36 | 86.14 ± 0.44 | 88.86 ± 0.11 | 90.35 ± 0.08 | 91.35 ± 0.24 |
| | Greedy k | 78.58 ± 0.91 | 86.37 ± 0.36 | 89.70 ± 0.33 | 91.71 ± 0.18 | 92.97 ± 0.10 |
| PruneFuse $p = 0.8$ | Least Conf. | 80.34 ± 0.39 | 88.00 ± 0.13 | 91.22 ± 0.07 | 92.89 ± 0.23 | 93.80 ± 0.23 |
| | Entropy | 79.61 ± 0.35 | 88.12 ± 0.00 | 90.94 ± 0.13 | 92.76 ± 0.14 | 93.54 ± 0.24 |
| | Random | 78.94 ± 0.49 | 86.20 ± 0.10 | 89.11 ± 0.34 | 90.50 ± 0.22 | 91.42 ± 0.23 |
| | Greedy k | 78.41 ± 0.76 | 85.90 ± 0.73 | 89.57 ± 0.51 | 91.38 ± 0.32 | 92.21± 0.22 |

(a) CIFAR-10 using ResNet-110 architecture.

| Method | Selection Metric | Label Budget ($b$) | | | | |
|---|---|---|---|---|---|---|
| | | 10% | 20% | 30% | 40% | 50% |
| Baseline $AL$ | Least Conf. | 38.61 ±0.32 | 54.47 ±0.56 | 61.46 ±0.25 | 65.96 ±0.48 | 68.91 ± 0.40 |
| | Entropy | 38.00 ± 0.99 | 54.71 ±0.83 | 60.82 ±0.15 | 66.19 ± 0.31 | 68.79 ± 0.50 |
| | Random | 37.88 ± 1.03 | 52.84 ±0.11 | 59.41 ±0.34 | 64.11 ± 0.11 | 67.22 ± 0.36 |
| | Greedy k | 37.41 ± 0.98 | 53.86 ±0.55 | 61.44 ±0.26 | 65.73 ± 0.50 | 68.17 ± 0.46 |
| PruneFuse $p = 0.5$ | Least Conf. | 41.42 ± 0.51 | 55.91 ± 0.36 | 62.43 ± 0.32 | 66.95 ± 0.20 | 69.79 ± 0.26 |
| | Entropy | 40.83 ± 0.59 | 56.29 ± 0.83 | 62.62 ± 0.45 | 66.91 ± 0.02 | 69.96 ± 0.39 |
| | Random | 40.36 ± 0.74 | 55.48 ± 0.25 | 61.14 ± 0.68 | 65.03 ± 0.42 | 67.85 ± 0.53 |
| | Greedy k | 41.22 ± 0.46 | 55.70 ± 0.54 | 62.27 ± 0.02 | 66.20 ± 0.14 | 68.86 ± 0.14 |
| PruneFuse $p = 0.6$ | Least Conf. | 38.52 ± 1.49 | 54.90 ± 0.32 | 61.50 ± 0.77 | 66.14 ± 0.68 | 69.03 ± 0.24 |
| | Entropy | 38.78 ± 1.35 | 53.13 ± 0.30 | 61.42 ± 0.14 | 65.62 ± 0.43 | 68.89 ± 0.09 |
| | Random | 40.24 ± 0.90 | 53.38 ± 0.68 | 59.93 ± 0.12 | 64.70 ± 0.15 | 66.62 ± 0.24 |
| | Greedy k | 39.99 ± 1.56 | 54.91 ± 2.23 | 61.04 ± 0.25 | 64.69 ± 0.63 | 67.60 ± 0.08 |
| PruneFuse $p = 0.7$ | Least Conf. | 37.83 ± 1.02 | 53.08 ± 0.25 | 61.41 ± 0.21 | 65.77 ± 0.43 | 68.03 ± 0.14 |
| | Entropy | 36.53 ± 0.97 | 52.97 ± 0.76 | 59.82 ± 0.63 | 64.97 ± 0.13 | 68.64 ± 0.54 |
| | Random | 39.46 ± 0.59 | 52.89 ± 0.77 | 59.92 ± 0.55 | 63.69 ± 0.25 | 66.30 ± 0.15 |
| | Greedy k | 40.44 ± 0.13 | 52.56 ± 0.28 | 59.83 ± 0.45 | 64.50 ± 0.29 | 66.99 ± 0.50 |
| PruneFuse $p = 0.8$ | Least Conf. | 38.33 ± 0.58 | 52.89 ± 0.49 | 60.08 ± 0.32 | 65.12 ± 0.60 | 68.06 ± 0.56 |
| | Entropy | 35.34 ± 0.98 | 51.88 ± 0.74 | 59.80 ± 0.82 | 64.58 ± 0.43 | 68.02 ± 0.17 |
| | Random | 38.22 ± 0.39 | 53.37 ± 0.72 | 59.84 ± 0.43 | 64.31 ± 0.33 | 67.23 ± 0.25 |
| | Greedy k | 37.72 ± 0.70 | 50.55 ± 1.79 | 57.39 ± 0.93 | 61.79 ± 0.53 | 65.21 ± 0.24 |

(b) CIFAR-100 using ResNet-110 architecture.

Table 7: **Performance Comparison** of Baseline and PruneFuse on CIFAR-10 and CIFAR-100 with ResNet-110 architecture. This table summarizes the test accuracy of final models (original in case of AL and Fused in PruneFuse) for various pruning ratios ($p$), labeling budgets ($b$), and data selection metrics.

## A.7 COMPARISON WITH SVP

Table 13 delineates a performance comparison of PruneFuse with SVP techniques, across various labeling budgets $b$ for the efficient training of a Target Model (ResNet-56). SVP employs a ResNet-20 as its data selector, with a model size of 0.26 M. In contrast, PruneFuse uses a 50% pruned ResNet-56, reducing its data selector size to 0.21 M. Performance metrics show that as the label budget increases from 10% to 50%, the PruneFuse consistently outperforms SVP across all label budgets. Specifically on the target model, PruneFuse initiates at an accuracy of 82.68% with a 10% label budget and peaks at 93.69% accuracy at a 50% budget, whereas SVP achieves 80.76% at 10% label budget and achieves

| Method | Selection Metric | Label Budget ($b$) | | | | |
|--------|------------------|------|------|------|------|------|
| | | 10% | 20% | 30% | 40% | 50% |
| Baseline AL | Least Conf. | $81.15 \pm 0.52$ | $89.4 \pm 0.27$ | $92.72 \pm 0.10$ | $94.09 \pm 0.14$ | $94.63 \pm 0.18$ |
| | Entropy | $80.99 \pm 0.44$ | $89.54 \pm 0.18$ | $92.45 \pm 0.16$ | $94.06 \pm 0.05$ | $94.49 \pm 0.09$ |
| | Random | $80.27 \pm 0.18$ | $87.00 \pm 0.08$ | $89.94 \pm 0.13$ | $91.57 \pm 0.09$ | $92.78 \pm 0.04$ |
| | Greedy k | $80.02 \pm 0.42$ | $88.33 \pm 0.47$ | $91.76 \pm 0.24$ | $93.39 \pm 0.22$ | $94.40 \pm 0.18$ |
| PruneFuse $p = 0.5$ | Least Conf. | $83.03 \pm 0.09$ | $90.30 \pm 0.06$ | $93.00 \pm 0.15$ | $94.41 \pm 0.08$ | $94.63 \pm 0.13$ |
| | Entropy | $82.64 \pm 0.22$ | $89.88 \pm 0.27$ | $93.08 \pm 0.25$ | $94.32 \pm 0.12$ | $94.90 \pm 0.13$ |
| | Random | $81.52 \pm 0.54$ | $87.84 \pm 0.15$ | $90.14 \pm 0.08$ | $91.94 \pm 0.18$ | $92.81 \pm 0.12$ |
| | Greedy k | $81.70 \pm 0.13$ | $88.75 \pm 0.33$ | $91.92 \pm 0.07$ | $93.64 \pm 0.04$ | $94.22 \pm 0.09$ |
| PruneFuse $p = 0.6$ | Least Conf. | $82.86 \pm 0.38$ | $90.22 \pm 0.18$ | $93.05 \pm 0.10$ | $94.27 \pm 0.06$ | $94.66 \pm 0.08$ |
| | Entropy | $82.23 \pm 0.39$ | $90.18 \pm 0.11$ | $92.91 \pm 0.15$ | $94.28 \pm 0.14$ | $94.66 \pm 0.14$ |
| | Random | $81.14 \pm 0.26$ | $87.51 \pm 0.26$ | $90.05 \pm 0.20$ | $91.82 \pm 0.22$ | $92.43 \pm 0.20$ |
| | Greedy k | $81.11 \pm 0.10$ | $88.41 \pm 0.18$ | $91.66 \pm 0.18$ | $92.94 \pm 0.12$ | $94.17 \pm 0.02$ |
| PruneFuse $p = 0.7$ | Least Conf. | $82.76 \pm 0.29$ | $89.89 \pm 0.17$ | $92.83 \pm 0.08$ | $94.10 \pm 0.08$ | $94.69 \pm 0.13$ |
| | Entropy | $82.59 \pm 0.69$ | $89.81 \pm 0.24$ | $92.77 \pm 0.07$ | $94.20 \pm 0.20$ | $94.74 \pm 0.02$ |
| | Random | $80.88 \pm 0.38$ | $87.54 \pm 0.26$ | $90.09 \pm 0.08$ | $91.57 \pm 0.26$ | $92.64 \pm 0.10$ |
| | Greedy k | $81.68 \pm 0.40$ | $88.36 \pm 0.56$ | $91.64 \pm 0.40$ | $93.02 \pm 0.42$ | $93.97 \pm 0.51$ |
| PruneFuse $p = 0.8$ | Least Conf. | $82.66 \pm 0.09$ | $89.78 \pm 0.27$ | $92.64 \pm 0.14$ | $94.08 \pm 0.10$ | $94.69 \pm 0.17$ |
| | Entropy | $82.01 \pm 0.88$ | $89.77 \pm 0.44$ | $92.65 \pm 0.09$ | $94.02 \pm 0.17$ | $94.60 \pm 0.18$ |
| | Random | $80.73 \pm 0.49$ | $87.43 \pm 0.44$ | $90.08 \pm 0.12$ | $91.40 \pm 0.07$ | $92.53 \pm 0.18$ |
| | Greedy k | $79.66 \pm 0.60$ | $87.56 \pm 0.12$ | $90.79 \pm 0.07$ | $92.30 \pm 0.12$ | $93.17 \pm 0.14$ |

(a) CIFAR-10 using ResNet-164 architecture.

| Method | Selection Metric | Label Budget ($b$) | | | | |
|--------|------------------|------|------|------|------|------|
| | | 10% | 20% | 30% | 40% | 50% |
| Baseline AL | Least Conf | $38.41 \pm 0.73$ | $51.39 \pm 0.30$ | $65.53 \pm 0.31$ | $70.07 \pm 0.17$ | $73.05 \pm 0.11$ |
| | Entropy | $36.65 \pm 0.76$ | $57.58 \pm 0.63$ | $64.98 \pm 0.30$ | $69.99 \pm 0.17$ | $72.90 \pm 0.15$ |
| | Random | $39.31 \pm 1.22$ | $57.53 \pm 0.26$ | $63.84 \pm 0.14$ | $67.75 \pm 0.14$ | $70.79 \pm 0.07$ |
| | Greedy k | $39.76 \pm 0.58$ | $57.40 \pm 0.20$ | $65.20 \pm 0.31$ | $69.25 \pm 0.40$ | $72.91 \pm 0.29$ |
| PruneFuse $p = 0.5$ | Least Conf | $42.88 \pm 1.11$ | $59.31 \pm 0.70$ | $66.95 \pm 0.30$ | $71.45 \pm 0.42$ | $74.32 \pm 0.58$ |
| | Entropy | $42.99 \pm 0.18$ | $59.32 \pm 1.25$ | $66.83 \pm 0.29$ | $71.18 \pm 0.40$ | $74.43 \pm 0.34$ |
| | Random | $43.72 \pm 1.05$ | $58.58 \pm 0.61$ | $64.93 \pm 0.43$ | $68.75 \pm 0.57$ | $71.63 \pm 0.40$ |
| | Greedy k | $43.61 \pm 0.91$ | $58.38 \pm 0.24$ | $66.04 \pm 0.21$ | $69.83 \pm 0.16$ | $73.10 \pm 0.39$ |
| PruneFuse $p = 0.6$ | Least Conf | $41.86 \pm 0.70$ | $58.97 \pm 0.50$ | $66.61 \pm 0.39$ | $70.59 \pm 0.11$ | $73.60 \pm 0.10$ |
| | Entropy | $42.43 \pm 0.95$ | $58.74 \pm 0.80$ | $65.97 \pm 0.39$ | $70.90 \pm 0.48$ | $73.70 \pm 0.09$ |
| | Random | $42.53 \pm 0.46$ | $58.33 \pm 0.42$ | $65.00 \pm 0.26$ | $68.55 \pm 0.30$ | $71.46 \pm 0.32$ |
| | Greedy k | $42.71 \pm 0.91$ | $58.41 \pm 0.18$ | $65.43 \pm 0.69$ | $69.57 \pm 0.14$ | $72.49 \pm 0.25$ |
| PruneFuse $p = 0.7$ | Least Conf | $42.00 \pm 0.20$ | $57.08 \pm 0.36$ | $66.41 \pm 0.30$ | $70.68 \pm 0.29$ | $73.63 \pm 0.29$ |
| | Entropy | $41.01 \pm 1.66$ | $57.45 \pm 0.50$ | $65.99 \pm 0.10$ | $70.07 \pm 0.54$ | $73.45 \pm 0.04$ |
| | Random | $42.76 \pm 1.00$ | $57.31 \pm 0.07$ | $64.12 \pm 0.57$ | $68.07 \pm 0.24$ | $70.88 \pm 0.25$ |
| | Greedy k | $42.42 \pm 0.32$ | $57.58 \pm 0.52$ | $65.18 \pm 0.51$ | $68.55 \pm 0.10$ | $71.89 \pm 0.16$ |
| PruneFuse $p = 0.8$ | Least Conf | $41.19 \pm 1.07$ | $57.98 \pm 9.70$ | $65.22 \pm 0.44$ | $70.38 \pm 0.22$ | $73.17 \pm 0.26$ |
| | Entropy | $39.78 \pm 1.16$ | $57.30 \pm 0.41$ | $65.19 \pm 0.63$ | $69.40 \pm 0.34$ | $72.82 \pm 0.03$ |
| | Random | $42.08 \pm 1.55$ | $57.23 \pm 0.47$ | $64.05 \pm 0.40$ | $67.85 \pm 0.19$ | $70.62 \pm 0.06$ |
| | Greedy k | $42.20 \pm 1.21$ | $57.42 \pm 0.50$ | $64.53 \pm 0.21$ | $68.01 \pm 0.40$ | $71.29 \pm 0.14$ |

(b) CIFAR-100 using ResNet-164 architecture.

Table 8: **Performance Comparison** of Baseline and PruneFuse on CIFAR-10 and CIFAR-100 with ResNet-164 architecture. This table summarizes the test accuracy of final models (original in case of AL and Fused in PruneFuse) for various pruning ratios ($p$), labeling budgets ($b$), and data selection metrics.

92.95% accuracy at 50%. Notably, while the data selector of PruneFuse achieves a lower accuracy of 90.31% at $b = 50\%$ compared to SVP's 91.61%, the target model utilizing PruneFuse-selected data attains a superior accuracy of 93.69%, relative to 92.95% for the SVP-selected data. This disparity underscores the distinct operational focus of the data selectors: PruneFuse's selector is optimized for enhancing the target model's performance, rather than its own accuracy. Fig. 4(a) and (b) show that target models ResNet-14 and ResNet-20, when trained with the data selectors of the PruneFuse achieve significantly higher accuracy while using significantly less number of parameters compared to SVP. These results indicate that the proposed approach does not require an additional architecture

| Method | Label Budget ($b$) | | | | |
|---|---|---|---|---|---|
| | 10% | 20% | 30% | 40% | 50% |
| Baseline ($AL$) | $14.86 \pm 0.11$ | $33.62 \pm 0.52$ | $43.96 \pm 0.22$ | $49.86 \pm 0.56$ | $54.65 \pm 0.38$ |
| PruneFuse ($p = 0.5$) | $18.71 \pm 0.21$ | $39.70 \pm 0.31$ | $47.41 \pm 0.20$ | $51.84 \pm 0.10$ | $55.89 \pm 1.21$ |
| PruneFuse ($p = 0.6$) | $19.25 \pm 0.72$ | $38.84 \pm 0.70$ | $47.02 \pm 0.30$ | $52.09 \pm 0.29$ | $55.29 \pm 0.28$ |
| PruneFuse ($p = 0.7$) | $18.32 \pm 0.95$ | $39.24 \pm 0.75$ | $46.45 \pm 0.58$ | $52.02 \pm 0.65$ | $55.63 \pm 0.55$ |
| PruneFuse ($p = 0.8$) | $18.34 \pm 0.93$ | $37.86 \pm 0.42$ | $47.15 \pm 0.31$ | $51.77 \pm 0.40$ | $55.18 \pm 0.50$ |

Table 9: **Performance Comparison** of Baseline and PruneFuse on Tiny ImageNet-200 with ResNet-50 architecture, including test accuracy and corresponding standard deviations. This table summarizes the test accuracy of final models (original in case of AL and Fused in PruneFuse) for various pruning ratios ($p$) and labeling budgets ($b$).

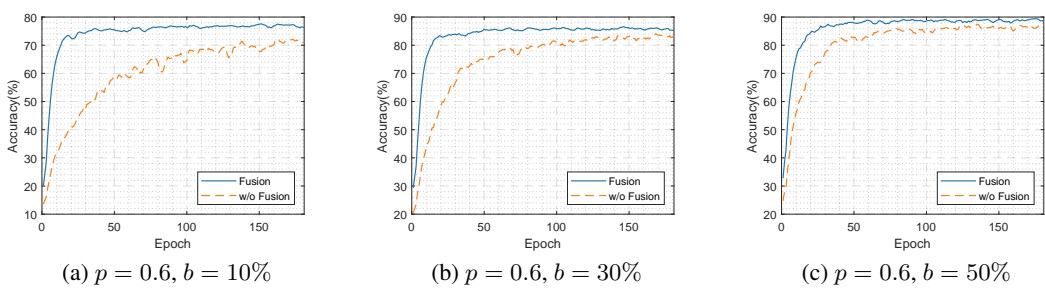

(a) $p = 0.6, b = 10\%$      (b) $p = 0.6, b = 30\%$      (c) $p = 0.6, b = 50\%$

Figure 7: **Ablation Study of Fusion on PruneFuse** ($p = 0.6$). Experiments are performed on ResNet-56 architecture with CIFAR-10.

for designing the data selector; it solely needs the target model (e.g. ResNet-14). In contrast, SVP necessitates both the target model (ResNet-14) and a smaller model (ResNet-8) that functions as a data selector.

Table 11 demonstrates the performance comparison of PruneFuse and SVP for small model architecture ResNet-20 on CIFAR-10. SVP achieves 91.88% performance accuracy by utilizing the data selector having 0.074 M parameters whereas PruneFuse outperforms SVP by achieving 92.29% accuracy with a data selector of 0.066 M parameters.

## A.8 ABLATION STUDY ON THE NUMBER OF SELECTED DATA POINTS ($k$)

Table 12 presents an ablation study analyzing the effect of varying $k$ on the performance of PruneFuse on CIFAR-10 using the ResNet-56 architecture and least confidence as the selection metric. The results demonstrate that the choice of $k$ significantly impacts the quality of data selection and the final performance of the model. As $k$ increases, the selected subset quality diminishes as can be seen by comparing performance of the target network when $b = 30\%$. This study highlights the importance of tuning $k$ to achieve an optimal trade-off between computational efficiency and model accuracy.

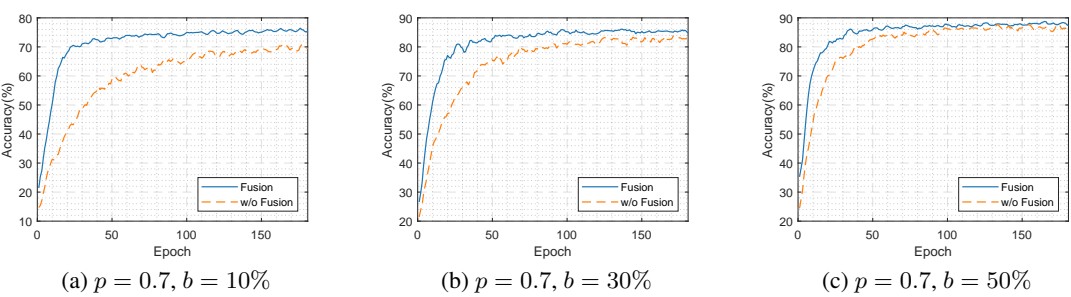

(a) $p = 0.7, b = 10\%$      (b) $p = 0.7, b = 30\%$      (c) $p = 0.7, b = 50\%$

Figure 8: **Ablation Study of Fusion on PruneFuse** ($p = 0.7$). Experiments are performed on ResNet-56 architecture with CIFAR-10.

| Method | Selection Metric | Label Budget ($b$) | | | | |
|---|---|---|---|---|---|---|
| | | 10% | 20% | 30% | 40% | 50% |
| Baseline
*AL* | Least Conf | 80.53 | 87.74 | 90.85 | 92.24 | 93.00 |
| | Entropy | 80.14 | 87.63 | 90.80 | 92.51 | 92.98 |
| | Random | 78.55 | 85.26 | 88.13 | 89.81 | 91.20 |
| | Greedy k | 79.63 | 86.46 | 90.09 | 91.90 | 92.80 |
| PruneFuse
$p = 0.5$
(without KD) | Least Conf | **81.08** | **88.71** | **91.24** | **92.68** | **93.46** |
| | Entropy | **80.80** | **88.08** | **90.98** | **92.74** | **93.43** |
| | Random | **80.11** | **85.78** | **88.81** | **90.20** | 91.10 |
| | Greedy k | **80.07** | **86.70** | 89.93 | 91.72 | 92.67 |
| PruneFuse
$p = 0.5$
(with KD) | Least Conf | 80.92 | 88.35 | 91.44 | 92.77 | 93.65 |
| | Entropy | 81.08 | 88.74 | 91.33 | 92.78 | 93.48 |
| | Random | 80.43 | 86.28 | 88.75 | 90.36 | 91.42 |
| | Greedy k | 79.85 | 86.96 | 90.20 | 91.82 | 92.89 |

(a) CIFAR-10 using ResNet-56 architecture.

| Method | Selection Metric | Label Budget ($b$) | | | | |
|---|---|---|---|---|---|---|
| | | 10% | 20% | 30% | 40% | 50% |
| Baseline
*AL* | Least Conf | 81.15 | 89.4 | 92.72 | 94.09 | 94.63 |
| | Entropy | 80.99 | 89.54 | 92.45 | 94.06 | 94.49 |
| | Random | 80.27 | 87.00 | 89.94 | 91.57 | 92.78 |
| | Greedy k | 80.02 | 88.33 | 91.76 | 93.39 | 94.40 |
| PruneFuse
$p = 0.5$
(without KD) | Least Conf | **83.82** | **90.26** | **93.15** | **94.34** | **94.90** |
| | Entropy | **82.72** | **90.42** | **93.18** | **94.68** | **95.00** |
| | Random | **81.94** | **88.04** | **90.37** | **91.93** | 92.67 |
| | Greedy k | **81.99** | **89.04** | **92.14** | **93.40** | **94.44** |
| PruneFuse
$p = 0.5$
(with KD) | Least Conf. | 83.03 | 90.30 | 93.00 | 94.41 | 94.63 |
| | Entropy | 82.64 | 89.88 | 93.08 | 94.32 | 94.90 |
| | Random | 81.52 | 87.84 | 90.14 | 91.94 | 92.81 |
| | Greedy k | 81.70 | 88.75 | 91.92 | 93.64 | 94.22 |

(b) CIFAR-10 using ResNet-164 architecture.

| Method | Selection Metric | Label Budget ($b$) | | | | |
|---|---|---|---|---|---|---|
| | | 10% | 20% | 30% | 40% | 50% |
| Baseline
*AL* | Least Conf | 35.99 | 52.99 | 59.29 | 63.68 | 66.72 |
| | Entropy | 37.57 | 52.64 | 58.87 | 63.97 | 66.78 |
| | Random | 37.06 | 51.62 | 58.77 | 62.05 | 64.63 |
| | Greedy k | 38.28 | 52.43 | 58.96 | 63.56 | 66.30 |
| PruneFuse
$p = 0.5$
(without KD) | Least Conf | 39.27 | **54.25** | 60.6 | 64.17 | 67.49 |
| | Entropy | 37.43 | 52.57 | 60.57 | 64.44 | 67.31 |
| | Random | **40.07** | 52.83 | 59.93 | 63.06 | 65.41 |
| | Greedy k | 39.25 | 52.43 | 59.94 | 63.94 | 66.56 |
| PruneFuse
$p = 0.5$
(with KD) | Least Conf | **40.26** | 53.90 | **60.80** | **64.98** | **67.87** |
| | Entropy | **38.59** | **54.01** | 60.52 | **64.83** | **67.67** |
| | Random | 39.43 | **54.60** | **60.13** | **63.91** | **66.02** |
| | Greedy k | **39.83** | **54.35** | **60.40** | **64.22** | **66.89** |

(c) CIFAR-100 using ResNet-56 architecture.

Table 10: Ablation Study of Knowledge Distillation on PruneFuse presented in a, b, and c with different architectures and datasets.

## A.9 IMPACT OF EARLY STOPPING ON PERFORMANCE

Table 14 explores the effect of utilizing an early stopping strategy alongside PruneFuse ($p = 0.5$) on CIFAR-10 with the ResNet-56 architecture. The results indicate that early stopping not only reduces training time of the fused model but also maintains comparable performance to fully trained models. This highlights the compatibility of PruneFuse with training efficiency techniques such as early stopping and showcases how the expedited convergence enabled by the fusion process further enhances its practicality, particularly in resource-constrained environments.

| Techniques | Model | Architecture | No. of Parameters (Million) | Label Budget ($b$) | | | | |
|---|---|---|---|---|---|---|---|---|
| | | | | 10% | 20% | 30% | 40% | 50% |
| SVP | Data Selector | ResNet-8 | 0.074 | 77.85 | 83.35 | 85.43 | 86.83 | 86.90 |
| | Target | ResNet-20 | 0.26 | 80.18 | 86.34 | 89.22 | 90.75 | 91.88 |
| PruneFuse | Data Selector | ResNet-20 ($p = 0.5$) | **0.066** | 76.58 | 83.41 | 85.83 | 87.07 | 88.06 |
| | Target | ResNet-20 | 0.26 | **80.25** | **87.57** | **90.20** | **91.70** | **92.29** |

Table 11: Comparison of SVP and PruneFuse on Small Models.

| Method | Label Budget ($b$) | | | | |
|---|---|---|---|---|---|
| | 15% | 30% | 45% | 60% | 75% |
| Baseline ($AL$) | 84.63 | 90.59 | 92.77 | 93.12 | 93.94 |
| PruneFuse ($p = 0.5$) | **85.80** | **91.13** | **93.72** | **93.84** | **94.10** |

(a) $k = 7.5K$.

| Method | Label Budget ($b$) | | | | |
|---|---|---|---|---|---|
| | 10% | 20% | 30% | 40% | 50% |
| Baseline ($AL$) | 80.53 | 87.74 | 90.85 | 92.24 | 93.00 |
| PruneFuse ($p = 0.5$) | **80.92** | **88.35** | **91.44** | **92.77** | **93.65** |

(b) $k = 5K$.

Table 12: **Ablation study of** $k$ on Cifar-10 using ResNet-56 architecture and least confidence as a selection matric.

## A.10   Performance Comparison Across Architectures and Datasets

In Table 15, we present the performance comparison of Baseline and PruneFuse across various architectures and datasets. These results demonstrate the adaptability of PruneFuse to different network architectures, including ResNet-18, ResNet-50, and Wide-ResNet (W-28-10), as well as datasets such as CIFAR-10, CIFAR-100, and ImageNet. The experiments confirm that PruneFuse consistently improves performance over the baseline, highlighting its generalizability and robustness across diverse scenarios.

## A.11   Performance at Lower Pruning Rates

Table 16 provides a performance comparison of Baseline and PruneFuse with a lower pruning rate of $p = 0.4$ on CIFAR-10 and CIFAR-100 using the ResNet-56 architecture. Least Confidence and Entropy were used as selection metrics for these experiments. The results show that even at a lower pruning rate, PruneFuse effectively selects high-quality data subsets, maintaining strong performance in both datasets. These findings validate the method's effectiveness across different pruning rates.

## A.12   Comparison with Recent Coreset Selection Techniques

Table 17 compares the performance of Baseline (Coreset Selection) and PruneFuse ($p = 0.5$) using various recent selection metrics, including Forgetting Events (Toneva et al., 2019), Moderate (Xia et al., 2022), and CSS (Zheng et al., 2022) on the CIFAR-10 dataset with the ResNet-56 architecture.

To incorporate these recent score metrics, which are specifically designed for coreset-based selection, we utilized the coreset task setup. In this setup, the network is first trained on the entire dataset to identify a representative subset of data (coreset) based on the selection metric. The accuracy of the target model trained on the selected coreset is then reported. The results demonstrate that PruneFuse seamlessly integrates with these advanced selection metrics, achieving competitive or superior performance compared to the baseline while maintaining computational efficiency. This highlights the versatility of PruneFuse in adapting to and enhancing existing coreset selection techniques.

## A.13   Effect of Various Pruning Strategies and Criterion

In Table 18, we evaluate the impact of different pruning techniques (e.g., static pruning, dynamic pruning) and pruning criteria (e.g., L2 norm, GroupNorm Importance, LAMP Importance [Fang et al. (2023)]) on the performance of PruneFuse ($p = 0.5$) on CIFAR-10 using the ResNet-56 architecture.

| Method | Model | Architecture | Params (Million) | Label Budget ($b$) | | | | |
|--------|-------|--------------|---------|-----|-----|-----|-----|-----|
| | | | | 10% | 20% | 30% | 40% | 50% |
| SVP | Data Selector | ResNet-20 | 0.26 | 81.07 | 86.51 | 89.77 | 91.08 | 91.61 |
| | Target | ResNet-56 | 0.85 | 80.76 | 87.31 | 90.77 | 92.59 | 92.95 |
| PruneFuse | Data Selector | ResNet-56 ($p = 0.5$)) | **0.21** | 78.62 | 84.92 | 88.17 | 89.93 | 90.31 |
| | Target | ResNet-56 | 0.85 | **82.68** | **88.97** | **91.63** | **93.24** | **93.69** |

Table 13: Comparison with SVP.

| Method | Epochs | Label Budget ($b$) | | | | |
|--------|--------|-----|-----|-----|-----|-----|
| | | 10% | 20% | 30% | 40% | 50% |
| Least Conf. | 181 | 80.92±0.409 | 88.35±0.327 | 91.44±0.148 | 92.77±0.026 | 93.65±0.141 |
| | 110 | 80.51±0.375 | 87.64±0.222 | 90.79±0.052 | 92.11±0.154 | 93.00±0.005 |
| Entropy | 181 | 81.08±0.155 | 88.74±0.103 | 91.33±0.045 | 92.78±0.045 | 93.48±0.042 |
| | 110 | 80.51±0.401 | 87.46±0.416 | 90.97±0.116 | 92.2±0.108 | 92.88±0.264 |
| Random | 181 | 80.43±0.273 | 86.28±0.367 | 88.75±0.17 | 90.36±0.022 | 91.42±0.125 |
| | 110 | 79.29±0.355 | 84.99±0.156 | 87.86±0.323 | 89.99±0.090 | 90.85±0.012 |
| Greedy k. | 181 | 79.85±0.676 | 86.96±0.385 | 90.20±0.164 | 91.82±0.136 | 92.89±0.144 |
| | 110 | 79.36±0.274 | 86.36±0.455 | 89.67±0.319 | 91.19±0.302 | 91.91±0.021 |

Table 14: **Performance Comparison** when Early Stopping strategy is utilized alongside PruneFuse ($p = 0.5$). Experiments are performed with Resnet-56 on CIFAR-10.

| Method | Label Budget ($b$) | | | | |
|--------|-----|-----|-----|-----|-----|
| | 10% | 20% | 30% | 40% | 50% |
| Baseline ($AL$) | 83.12 | 90.07 | 92.71 | 94.07 | 94.81 |
| PruneFuse ($p = 0.5$) | **83.29** | **90.56** | **93.17** | **94.56** | **95.08** |

(a) ResNet-18 architecture on CIFAR-10.

| Method | Label Budget ($b$) | | | | |
|--------|-----|-----|-----|-----|-----|
| | 10% | 20% | 30% | 40% | 50% |
| Baseline ($AL$) | 84.74 | 91.48 | 94.17 | 95.24 | 95.75 |
| PruneFuse ($p = 0.5$) | **85.65** | **92.27** | **94.65** | **95.73** | **96.24** |

(b) Wide-ResNet architecture on CIFAR-10.

| Method | Label Budget ($b$) | | | | |
|--------|-----|-----|-----|-----|-----|
| | 10% | 20% | 30% | 40% | 50% |
| Baseline ($AL$) | 52.97 | 64.52 | 69.30 | 71.98 | 73.56 |
| PruneFuse ($p = 0.5$) | **55.03** | **65.12** | **69.72** | **72.07** | **73.86** |

(c) ResNet-50 architecture on ImageNet-1K.

Table 15: Performance Comparison of Baseline and PruneFuse presented in a, b, and b with different architectures and datasets.

Static pruning involves pruning the entire network at once at the start of training, whereas dynamic pruning incrementally prunes the network in multiple steps during training. In our implementation of dynamic pruning, the network is pruned in five steps over the course of 20 epochs.

The results demonstrate that PruneFuse is highly adaptable to various pruning strategies, consistently maintaining strong performance in data selection tasks. This flexibility underscores the robustness of the framework across different pruning approaches and criteria.

| Method | Selection Metric | Label Budget ($b$) | | | | |
|---|---|---|---|---|---|---|
| | | 10% | 20% | 30% | 40% | 50% |
| Baseline ($AL$) | Least Confidence | 80.53 | 87.74 | 90.85 | 92.24 | 93.00 |
| | Entropy | 80.14 | 87.63 | 90.80 | 92.51 | 92.98 |
| PruneFuse ($p = 0.4$) | Least Confidence | **81.12** | **88.16** | **91.35** | **92.89** | **93.20** |
| | Entropy | **80.94** | **88.27** | **91.09** | **92.73** | **93.38** |

(a) CIFAR-10

| Method | Selection Metric | Label Budget ($b$) | | | | |
|---|---|---|---|---|---|---|
| | | 10% | 20% | 30% | 40% | 50% |
| Baseline ($AL$) | Least Confidence | 35.99 | 52.99 | 59.29 | 63.68 | 66.72 |
| | Entropy | 37.57 | 52.64 | 58.87 | 63.97 | 66.78 |
| PruneFuse ($p = 0.4$) | Least Confidence | **38.73** | **54.35** | **60.75** | **64.80** | **67.08** |
| | Entropy | **38.35** | **54.19** | **60.79** | **65.00** | **67.47** |

(b) CIFAR-100

Table 16: **Performance Comparison of Baseline and PruneFuse**($p = 0.4$) on Cifar-10 and Cifar-100 using ResNet-56 architecture.

| Method | Selection Metric | Data Selector's Params | Target Model's Params | Accuracy ($b = 25\%$) |
|---|---|---|---|---|
| **Baseline** | Entropy | 0.85 Million | 0.85 Million | 86.13 |
| | Least Confidence | | | 86.50 |
| | Forgetting Events | | | 86.01 |
| | Moderate | | | 86.27 |
| | CSS | | | 87.21 |
| **PruneFuse** | Entropy | 0.21 Million | 0.85 Million | **86.71** |
| | Least Confidence | | | **86.68** |
| | Forgetting Events | | | **87.84** |
| | Moderate | | | **87.63** |
| | CSS | | | **88.85** |

Table 17: **Performance Comparison of Baseline (Coreset) and PruneFuse ($p = 0.5$) for Various selection metrics** including Forgetting Events (Toneva et al., 2019), Moderate (Xia et al., 2022), and CSS (Zheng et al., 2022) on Cifar-10 dataset using ResNet-56 architecture.

### A.14 RUNTIME COMPARISON OF DATA SELECTOR NETWORKS AND DETAILED BREAKDOWN OF THE TRAINING RUNTIME FOR EACH COMPONENT OF PRUNEFUSE

Table 19 compares the training runtimes of the data selector network (pruned network for PruneFuse and dense network for the baseline) across various network architectures. The reported times correspond to the training phase of the data selector network prior to the final selection of the subset (at $b = 50\%$, label budget). Note that the variation in runtimes across different datasets is due to the experiments being conducted on different servers, each equipped with specific GPUs (e.g., 2080Ti, 3090, or A100). The results show that PruneFuse significantly reduces training time due to the efficiency of the pruned network as compared to baseline, making it well suited for resource-constrained environments.

Table 20 provides a detailed breakdown of the training run time for each component of PruneFuse, including the data selector training time, the selection time, and the target network training time. These measurements offer a comprehensive view of the computational requirements of PruneFuse, demonstrating its efficiency compared to the baseline methods. The breakdown highlights that the pruned network and the fusion process contribute to significant computational savings without compromising performance.

| Method | Pruning Criteria | Label Budget ($b$) | | | | |
|---|---|---|---|---|---|---|
| | | 10% | 20% | 30% | 40% | 50% |
| **Baseline (AL)** | - | 80.53 | 87.74 | 90.85 | 92.24 | 93.00 |
| **PruneFuse** (Dynamic Pruning) | Magnitude Imp. | 79.73 | 87.16 | **91.08** | 92.29 | 93.19 |
| | GroupNorm Imp. | 80.10 | 88.25 | 91.01 | 92.25 | 93.74 |
| | LAMP Imp. | 81.51 | 87.45 | 90.64 | **92.41** | 93.25 |
| **PruneFuse** (Static Pruning) | Magnitude Imp. | 80.92 | 88.35 | 91.44 | 92.77 | 93.65 |
| | GroupNorm Imp. | 80.84 | 88.20 | 91.19 | 93.01 | 93.03 |
| | LAMP Imp. | 81.10 | 88.37 | 91.32 | 93.02 | 93.08 |
| **PruneFuse_V2** (Static Pruning) | Magnitude Imp. | 81.23 | 88.52 | 91.76 | 93.15 | 93.78 |
| | GroupNorm Imp. | 81.09 | 88.77 | 91.77 | 93.19 | 93.68 |
| | LAMP Imp. | 81.86 | 88.51 | 92.10 | 93.02 | 93.63 |

Table 18: **Effect of different Pruning Techniques and Pruning Criterion** on PruneFuse ($p = 5$) on Cifar-10 dataset with ResNet-56 architecture.

| Datasets | Data Selectors (Selection Models) | Training Runime (Minutes) |
|---|---|---|
| **CIFAR-10** | ResNet-56 (Baseline) | 127.67 |
| | ResNet-56 (PruneFuse ($p = 0.5$)) | **72.55** |
| | ResNet-56 (PruneFuse ($p = 0.8$)) | **67.23** |
| | ResNet-18 (Baseline) | 85.68 |
| | ResNet-18 (PruneFuse ($p = 0.5$)) | **61.15** |
| | Wide ResNet (Baseline) | 122.43 |
| | Wide ResNet (PruneFuse ($p = 0.5$)) | **75.48** |
| **CIFAR-100** | ResNet-164 (Baseline) | 129.23 |
| | ResNet-164 (PruneFuse ($p = 0.5$)) | **83.52** |
| | ResNet-164 (PruneFuse ($p = 0.8$)) | **78.55** |
| | ResNet-110 (Baseline) | 95.80 |
| | ResNet-110 (PruneFuse ($p = 0.5$)) | **80.42** |
| | ResNet-110 (PruneFuse ($p = 0.8$)) | **69.50** |
| **TinyImagenet-200** | ResNet-50 (Baseline) | 248.48 |
| | ResNet-50 (PruneFuse ($p = 0.5$)) | **147.47** |
| | ResNet-50 (PruneFuse ($p = 0.8$)) | **94.42** |
| **ImageNet-1K** | Resnet-50 (Baseline) | 2081.3 |
| | ResNet-50 (PruneFuse ($p = 0.5$)) | **951.17** |

Table 19: **Training Runtime** of data selector network i.e. pruned network in the case of PruneFuse and dense network for baseline, for various network architectures. The reported time is the training time when the network is trained before selecting final subset of the data ($b = 50\%$).

| Datasets | Label Budget ($b$) | Data Selectors (Training Time) (Minutes) | Data Selection Time (Minutes) | Target Model (Training Time) (Minutes) |
|---|---|---|---|---|
| **Baseline (AL)** | 10% | 48.80 | 4.43 | 48.80 |
| | 20% | 99.23 | 3.50 | 99.23 |
| | 30% | 145.32 | 3.15 | 145.32 |
| | 40% | 195.38 | 2.72 | 195.38 |
| | 50% | 248.48 | 2.38 | 248.48 |
| **PruneFuse** | 10% | **32.17** | **1.57** | 49.50 |
| | 20% | **61.70** | **1.67** | 99.99 |
| | 30% | **88.53** | **1.52** | 146.25 |
| | 40% | **117.10** | **1.37** | 196.28 |
| | 50% | **147.47** | **1.18** | 249.58 |

Table 20: **Detailed Training time of Baseline and PruneFuse(**$p = 0.5$**)** for TinyImageNet-200 for Resnet-50 using Least Confidence as selection metric.

