# OpenReview forum: "PruneFuse: Efficient Data Selection via Weight Pruning and Network Fusion"
_ICLR.cc/2025/Conference — Submitted to ICLR 2025_

### Official Review · Reviewer_vVDM · 2024-10-28

**Soundness:** 2
**Presentation:** 2
**Contribution:** 2
**Rating:** 5
**Confidence:** 4

**Summary:**

This paper proposes a novel strategy, PruneFuse, for efficient data selection in active learning setting. It employs model pruning to reduce the complexity of neural networks while preserving the accuracy. PruneFuse uses a pruned model for data selection and employs it to train the final model through a fusion process which can accelerate convergence and improve the generalization of the final model.

**Strengths:**

The method is to train a pruned model and fuse it with the original one to get a large mode while saving time, which is useful in continuous large model training.

**Weaknesses:**

- I'm a bit confused about Figure 2, which seems somewhat too idealized. Using actual training trajectories would make this figure clearer and more convincing.
- The initialization process seems quite random and seems to be unstable.
- The Baseline in Table 2 doesn't correspond to each other between Tsync = 1 and Tsync = 2 when label budget is 50%, which also unmatch the data in Table 1.
- The motivation for using pruned networks is not very clear, as it seems other teacher-student models with similar structures can achieve comparable effects.

**Questions:**

- My understanding is that the final model output by Prunefuse is a model with the same parameter number as the original, so I'm a little confused about the parameter count for Prunefuse in Table 1,  The changes in parameter counts don't correspond proportionally to the pruning rates.
- The performance of Prunefuse shown in Table 1 seems unstable and lacks a clear pattern, have the authors investigated potential reasons for this?
- Also, since Prunefuse achieved good results at p=50%, did the authors try experiments with p=40% or lower?
- I notice that the pruning method used in this paper is static structural pruning. While there are many dynamic pruning methods available nowadays. I wonder if the authors have tried any of these methods?

---

> ### Author Response · Authors · 2024-11-25
>
> We appreciate the reviewers’ feedback and the opportunity to address their concerns. We have responded to each comment below to provide additional clarity.
>
> ---
>
> >_1.	Confusion about Figure 2. Using actual trajectories would make this figure clearer._
>
> Figure 2 provides a conceptual visualization of the proposed framework, illustrating how pruning and fusion reshape the optimization dynamics and improve convergence. It is not intended to depict actual training trajectories. For empirical evidence, we direct the reviewer to Figure 5, where the training trajectories are shown. The results in Figure 5 clearly demonstrate that the proposed model fusion achieves faster convergence and better accuracy due to improved initialization of the network.
>
> ---
>
> >_2.	The initialization process seems to be unstable._
>
> Pruning from Scratch [1] demonstrates that pruning at initialization not only reduces training time but also provides robust solutions by enabling the exploration of sparse architectures that generalize well. In our experiments, we observed consistent and stable training behavior across multiple runs with different random seeds, confirming that the initialization process does not introduce instability. Specifically, the pruned networks exhibit predictable performance trends: as the pruning rate increases, the accuracy of the network decreases proportionally. Furthermore, we observe that data selection quality and the overall performance of PruneFuse improve when the pruned network retains more parameters, underscoring the robustness of our initialization process. These empirical results and theoretical insights from prior work validate the stability of the initialization process used in our method.
>
> ---
>
> >_3.	The Baseline in Table 2 doesn't correspond to each other between Tsync = 1 and Tsync = 2 when label budget is 50%, which also unmatch the data in Table 1._
>
> Thank you for pointing this out. We made a typo in Table 2 (Tysnc=1 for 50% budget), the value should be 93.61%. We have fixed it in the revised manuscript. Since Tsync is a parameter specific to PruneFuse V2, the baseline results remain the same for Tsync=1 and Tsync=2.
>
> Additionally, as Table 1 reports results for PruneFuse V1 and Table 2 focuses on PruneFuse V2, the baseline methodology was slightly adjusted for fairness. Specifically, in Table 2 when compared against the PruneFuseV2, which uses the trained fused model to guide the data selector, we modified the baseline to continue retraining the network from the previous round rather than reinitializing it (also mentioned in the results section 5.2). This adjustment led to slight differences in baseline accuracy and ensures a consistent and fair comparison within the context of PruneFuse V2.
>
> ---
>
> >_4.	The motivation for using pruned networks is not very clear, as it seems other teacher-student models with similar structures can achieve comparable effects._
>
> The primary motivation for our approach is to enhance the efficiency of active learning pipelines, particularly in resource-constrained environments, while maintaining a generic and adaptable framework. Handcrafted teacher-student models, often require significant manual effort to design and are not easily adaptable to varying computational or task-specific demands. As demonstrated in comparison with SVP (Fig. 4 in the main paper and Table 13 in the Supplementary Materials), such models may not always achieve optimal compatibility or performance.
>
> Additionally, the student models used as surrogates in traditional teacher-student frameworks are typically being discarded after data selection, as there is no systematic way to integrate the insights gained during their training back into the teacher model. In contrast, our approach provides a systematic and generic method for designing data selectors through pruning and fusion, allowing us to utilize the knowledge from the trained selector models to improve the accuracy and efficiency of the final models. The proposed pipeline is scalable, adaptable, and applicable across diverse scenarios and computational settings, addressing the challenges and inefficiencies associated with teacher-student models while providing a more robust and reusable framework.

---

> > ### Author Response · Authors · 2024-11-25
> >
> > >_Q1. My understanding is that the final model output by Prunefuse is a model with the same parameter number as the original, so I'm a little confused about the parameter count for Prunefuse in Table 1, The changes in parameter counts don't correspond proportionally to the pruning rates._
> >
> > The parameters count in Table 1 correspond to the number of parameters of the data selector network (Pruned Network in our case and the dense network in case of baseline). We have updated the text surrounding Table 1 to further clarify this.
> >
> > Regarding the relationship between pruning rates and parameter counts, the changes do correspond proportionally. For instance, at a 0.5 pruning ratio for ResNet-56, the number of channels are halved from \(\{16, 32, 64\}\) in the original network to \(\{8, 16, 32\}\) in the pruned network. This reduction decreases the total number of parameters by approximately 75% (0.85M -> 0.21M). We hope this resolves any further confusion.
> >
> > ---
> >
> > >_Q2. The performance of Prunefuse shown in Table 1 seems unstable and lacks a clear pattern, have the authors investigated potential reasons for this?_
> >
> > We would like to clarify that the general pattern observed aligns with expectations: as the pruning ratio increases, the quality of the selected data decreases, resulting in lower accuracy during training. This trend is consistent with the hypothesis that higher pruning ratios reduce the capacity of the pruned network, leading to inferior data selection quality.
> >
> > For smaller datasets like CIFAR-10, where the number of data points becomes significantly reduced under high pruning regimes, accuracy fluctuations can appear more pronounced, which may give the impression of an unclear pattern. However, for larger datasets such as Tiny-ImageNet and ImageNet, the pattern becomes more consistent and pronounced due to the greater abundance of data, even at higher pruning rates.
> >
> > ---
> >
> > >_Q3. Since Prunefuse achieved good results at p=50%, did the authors try experiments with p=40% or lower?_
> >
> > Yes, we conducted experiments with \(p=40\%\), and the results are provided in Table 16 of the Supplementary Materials. These results exhibit a similar pattern, further validating the consistency and effectiveness of PruneFuse across varying pruning rates.
> >
> > ---
> >
> > >_Q4. Pruning method used in this paper is static structural pruning. While there are many dynamic pruning methods available nowadays. I wonder if the authors have tried any of these methods?_
> >
> > While the paper primarily focuses on static structural pruning, we have explored various pruning strategies, including dynamic pruning methods and pruning with alternative metrics. The results of these experiments, detailed in Table 18 of the Supplementary Materials, demonstrate the adaptability of PruneFuse to different pruning approaches while maintaining strong performance in both data selection and computational efficiency.
> >
> > We hope that our responses and the modifications in the revised version of the paper adequately address all concerns. Should there be any additional questions or points requiring further elaboration, we would be happy to provide clarification. We kindly request you to reconsider the contributions and impact of our work in light of the revisions and detailed explanations provided.
> >
> >
> > ***
> >
> > References
> >
> > *[1] Wang, Yulong, et al. "Pruning from scratch." Proceedings of the AAAI conference on artificial intelligence. Vol. 34. No. 07. 2020.*

---

> > > ### Comment · Reviewer_vVDM · 2024-11-26
> > >
> > > Thank you for the detailed responses. I have two additional concerns regarding the pruning ratio selection:
> > >
> > > 1. Given that p=0.4 shows improvements over p=0.5 in certain scenarios, I am curious about the performance with even lower pruning ratios. Could the authors provide results or insights for p<0.4? This would help establish a more complete understanding of the relationship between pruning ratio and model performance.
> > >
> > > 2. The selection of the optimal pruning ratio appears to be a critical factor, yet the current approach seems to require empirical testing for each scenario. Could the authors address:
> > >    - How to determine appropriate pruning ratios for different combinations of datasets, model architectures, and data budgets without exhaustive search?
> > >    - Whether there exist any theoretical guidelines or heuristics for pruning ratio selection?
> > >    - How to balance the trade-off between finding optimal pruning ratios and the method's primary goal of improving efficiency?
> > >
> > > This practical aspect seems particularly important, as conducting extensive experiments to determine optimal pruning ratios for each new scenario would contradict the method's efficiency objectives.

---

> > > > ### Author Response · Authors · 2024-12-01
> > > >
> > > > We thank the reviewer for their follow-up questions. Below, we address each concern in detail:
> > > >
> > > > >_1. Insights for lower pruning ratios_
> > > >
> > > > We performed extensive experimentation with various lower pruning ratios across different datasets and architectures, and observed that lower pruning ratios (e.g., $p=0.3$ or $p=0.2$) tend to achieve slightly better performance compared to higher ratios. This is because a larger portion of the network is trained, resulting in better data selection and improved model fusion. However, it is important to note that these benefits come at the cost of increased computational requirements for training the pruned network and performing data selection. This trade-off becomes particularly significant in resource-constrained environments. Comparatively, the range $p=0.5$ to $p=0.7$ consistently offers a robust trade-off between efficiency and performance. Within this range, significant reductions in computational costs are observed without sacrificing substantial accuracy, making it the most practical choice for diverse applications. To further clarify this trade-off, we will include detailed experimental results for lower pruning ratios in the updated version of the paper.
> > > >
> > > > >_2.	The selection of the optimal pruning ratio appears to be a critical factor, yet the current approach seems to require empirical testing for each scenario._
> > > >
> > > > _a). Determining Appropriate Pruning Ratios._
> > > >
> > > > Our experiments (as shown in Table 1) demonstrate that pruning ratios within the range $p=0.5$ to $p=0.7$ consistently achieve an optimal trade-off between computational efficiency and generalization performance. This range has been validated across diverse datasets and architectures, making it a reliable default setting for practitioners without the need for exhaustive empirical testing.
> > > >
> > > > _b). Heuristics for Pruning Ratio Selection._
> > > >
> > > > Our findings align closely with prior works, such as the Lottery Ticket Hypothesis [1] and Pruning from Scratch [2]. [1] demonstrates that sparse subnetworks retaining as little as 20% of the original network's weights can achieve comparable or superior performance to the original dense network, highlighting the feasibility of significant pruning without compromising accuracy. Similarly, [2] shows that pruning ratios up to $p = 0.7$ yield acceptable results, with $p = 0.5$ matching the performance of the dense network. These works together with our findings, suggest the range for best trade-off between performance and efficiency i.e., lower pruning ratios e.g., $p = 0.5$ yield better generalization, whereas higher ratios like $p = 0.7$ are suited for maximizing computational efficiency.
> > > >
> > > > _c). Balancing the Trade-Off Between Efficiency and Performance._
> > > >
> > > > The selection of pruning ratios inherently involves a trade-off between computational savings and model accuracy. Ratios closer to $p = 0.7$ significantly reduce parameter counts and runtime, making them ideal for resource-constrained environments or large-scale datasets. Conversely, ratios $p <= 0.5$ maintain strong generalization and better accuracy, particularly for smaller datasets or scenarios where model performance is critical. Overall, pruning ratios within the range of $p = 0.5$ to $p=0.7$ consistently achieve this balance, enabling PruneFuse to remain both effective and efficient without requiring extensive empirical tuning.
> > > >
> > > > ---
> > > > References
> > > >
> > > > *[1] Frankle, Jonathan, and Michael Carbin. "The Lottery Ticket Hypothesis: Finding Sparse, Trainable Neural Networks." International Conference on Learning Representations. 2018.*
> > > >
> > > > *[2] Wang, Yulong, et al. "Pruning from scratch." Proceedings of the AAAI conference on artificial intelligence. Vol. 34. No. 07. 2020.*

---

### Official Review · Reviewer_EpRF · 2024-10-29

**Soundness:** 1
**Presentation:** 3
**Contribution:** 3
**Rating:** 3
**Confidence:** 4

**Summary:**

This paper proposes PruneFuse to address the issue that traditional methods often face high computational costs. PruneFuse operates in two stages: pruning a network and training the pruned network with knowledge distillation, which will be used to select data. The trained network is fused with the original network and fine-tuned on the selected datasets.

**Strengths:**

1. The proposed method mitigates the need for continuous large model training prior to data selection.
2. This work introduces a new pipeline, fusing the trained network with the original untrained model.
3. Experiments are conducted on CIFAR-10/100, Tiny-ImageNet, and ImageNet-1k.

**Weaknesses:**

1. As emphasized in the Abstract, the primary motivation of this work is to address the high computational costs of traditional methods. However, IMHO, neural network pruning typically has high training costs. Meanwhile, the pruned network is also trained for fusion, which increases the training costs. So, I doubt the actual computational costs of the proposed method. Can the proposed method really obtain lower computational costs?
2. As far as I know, many selection methods have relatively low computational costs, such as Moderate [1], CCS[2]. Without reporting and comparing the training costs of each module of the proposed method, I don’t think this contribution is significant.
3. How is the Figure 2 drawer? Did the authors track the gradient direction and values? What is the meaning of different colors? More clarifications are needed.
4. The models experience pruning and then training. What if directly using pretrained models? This can denote the sample scores, as the forward pass can be finished very efficiently.
5. In experiments, authors only compare with several different score functions, while many state-of-the-art methods are not compared. I have doubts about the practical performance of the proposed methods. I recommend discussing and comparing with more advanced existing STOA methods, such as [1-6].
6. I highly recommend that the authors report the training costs of each specific module of the proposed method. Especially the overall training costs to obtain a selected model and obtain the trained model on the selected datasets.
7. Since the model is fused with a pretrained model (which is trained on the whole dataset), the knowledge acquired from the entire dataset is introduced. Therefore, it is unfair to compare directly with another selection method. For a fair comparison, authors are suggested to use the fused models to fine-tune different selected datasets from different baselines. This could significantly enhance the effectiveness of the proposed method.
8. Why do the experiments use some seldomly used architecture, such as ResNet-56, ResNet-14, ResNet-8, and ResNet-20? Authors are suggested to evaluate using more widely used models, such as ResNet-50, ResNet-18, Wide-ResNet, ViT, etc.

[1] Xia, Xiaobo, et al. "Moderate coreset: A universal method of data selection for real-world data-efficient deep learning." The Eleventh International Conference on Learning Representations. 2022.
[2] Zheng, Haizhong, et al. "Coverage-centric coreset selection for high pruning rates." arXiv preprint arXiv:2210.15809 (2022).
[3] Yang, Shuo, et al. "Dataset pruning: Reducing training data by examining generalization influence." arXiv preprint arXiv:2205.09329 (2022).
[4] Maharana, Adyasha, Prateek Yadav, and Mohit Bansal. "D2 pruning: Message passing for balancing diversity and difficulty in data pruning." arXiv preprint arXiv:2310.07931 (2023).
[5] Yang, Suorong, et al. "Not All Data Matters: An End-to-End Adaptive Dataset Pruning Framework for Enhancing Model Performance and Efficiency." arXiv preprint arXiv:2312.05599 (2023).
[6] Tan, Haoru, et al. "Data pruning via moving-one-sample-out." Advances in Neural Information Processing Systems 36 (2024).

**Questions:**

Please see weakness.

---

> ### Author Response · Authors · 2024-11-25
>
> Thank you for the thorough evaluation of our paper. We value the detailed feedback and have provided clarifications to each of the comments below.
>
> ---
>
> >_1.	Concerns regarding the computational costs when neural network pruning typically has high training costs. Also, the pruned network is also trained for fusion, which increases the training costs._
>
> The pruning process is performed once before training, involving a single computation of L2 norms and sorting, with a complexity of $O(P$ $log P)$. The fusion process, which integrates the weights from the trained pruned network $\theta_p^*$ into the original network $\theta$, is a lightweight operation with a complexity of $O(P)$, introducing negligible overhead. The detailed complexity analysis is provided in the Supplementary Materials section A.1.
> The primary computational effort lies in training the pruned network for data selection, which, particularly in the context of active learning, is performed over multiple rounds. In PruneFuse, the reduced size of the pruned network ensures significant efficiency compared to training the full network in each round, as demonstrated in Figure 3 of the main paper. Additionally, the fusion process accelerates convergence during the subsequent training of the full model, as illustrated in Figure 5.
> Furthermore, we have now included runtime comparison of PruneFuse versus the baseline in detail in Table 19 and 20 of the Supplementary Materials, further underscoring the practical efficiency of our approach. Together, these results reinforce the computational effectiveness and scalability of the proposed methodology.
>
> ---
>
> >_2.	Selection methods like Moderate and CCS have low computation costs. Report and compare the training costs of each module._
>
> The suggested techniques, Moderate [1] (which selects data points closer to the distance median from a class center) and CCS [2] (which improves data coverage for coreset selection), are primarily scoring strategies rather than computational optimizations for the data selection pipeline. Both methods require training the same model on which their scoring is based, resulting in computational costs comparable to other data selection strategies.
> That being said, these scoring strategies can be seamlessly integrated into the proposed PruneFuse pipeline. For a comprehensive evaluation, we have incorporated Moderate and CCS into our framework and present the results in Table 17 of the Supplementary Materials. The experiments demonstrate that while these strategies provide alternative scoring mechanisms, the computational efficiency and overall performance of PruneFuse remain superior due to the reduced size of the pruned network and the efficiency of the fusion process. These findings highlight the adaptability of PruneFuse and its ability to integrate diverse scoring strategies while maintaining its computational and performance advantages.
>
> ---
>
> >_3.	How is figure 2 drawn? Clarify the meaning of different colors._
>
> Figure 2 conceptually illustrates the evolution of training trajectories under the proposed framework. The contours represent the loss landscape, with colors transitioning from red (higher loss) to blue (lower loss). Subfigure 2a shows the trajectory of the original network $\theta$ in its unmodified loss landscape. After pruning, the landscape is tailored, as shown in 2b going from yellow (high loss) to blue (lower loss), simplifying the optimization process for the pruned network $\theta_p$, which converges to an optimal point denoted as $\theta_p^*$. Subfigure 2c demonstrates the refined trajectory of the fused model $\theta_F$, which benefits from the initialization provided by $\theta_p^*$, achieving a superior trajectory and improved convergence in the original landscape.
>
> This conceptual illustration aligns with the empirical results in Figure 5, where the faster convergence of the fused model $\theta_F$ is clearly demonstrated. Together, these figures emphasize the role of pruning in reshaping the optimization dynamics and the advantages introduced by the fusion process.

---

> > ### Author Response · Authors · 2024-11-25
> >
> > >_4.	Directly using pretrained models?_
> >
> > While pretrained models can be used for sample scoring via forward passes, they often perform suboptimally compared to the standard pool-based active learning pipeline due to differences between the pretrained model's learned distribution and the characteristics of the target dataset. Recent works, such as [7] and [8], show that training a data selector directly on the target dataset remains the most widely used and effective pipeline in active learning.
> > PruneFuse aligns with this approach by training a pruned model on the target dataset to ensure alignment with dataset-specific characteristics, enabling superior sample selection. At the same time, we also demonstrate the possibility of utilizing a pretrained network to generate the pruned network for subsequent data selection. In PruneFuse V2 (Section 4.6), pruning is performed on a trained fused model to create a refined pruned network, which is then used to enhance the data selection process. This highlights the adaptability and robustness of PruneFuse across diverse scenarios.
> >
> > ---
> >
> > >_5.	State of the art score functions._
> >
> > For our analysis, we utilized the most commonly known score functions to establish the baseline performance of our framework. However, to demonstrate the compatibility of PruneFuse with more advanced scoring strategies, we conducted additional experiments incorporating several recent SOTA score functions. These results are provided in Table 17 of the Supplementary Materials in the revised paper.
> > Our findings show that PruneFuse integrates seamlessly with these advanced score functions, maintaining its computational efficiency while achieving comparable or superior performance in data selection tasks. These experiments further validate the adaptability and practical utility of our proposed method.
> >
> > ---
> >
> > >_6.	Report the training costs of each specific module._
> >
> > We have conducted a detailed training cost analysis and compared it with baseline methods, as presented in Supplementary Materials Section A.1. Additionally, we now provide a comprehensive runtime breakdown of PruneFuse in Table 19 and Table 20 of the Supplementary Materials. These tables detail the training costs of each specific module, including the data selector, selection process, and target network training, further demonstrating the efficiency of our approach.
> >
> > ---
> >
> > >_7.	Due to model fusion with pretrained model, the knowledge from entire dataset is introduced._
> >
> > We clarify that the fusion process in PruneFuse is performed using the trained pruned model, which is trained only on the selected subset of the dataset based on the scoring criteria, and not on the entire dataset. This ensures that the knowledge introduced during fusion is derived exclusively from the selected subset, maintaining fairness in comparisons with other data selection methods.
> >
> > ---
> >
> > >_8.	Why experiments are conducted with ResNet-56, ResNet-14, ResNet-8 and ResNet-20 as compared to widely used ResNet-50, ResNet-18, Wide-Resnet, ViT etc_
> >
> > We used ResNet-56 and similar variants in our experiments as they are computationally efficient and well-suited for smaller datasets like CIFAR-10. For larger datasets such as ImageNet, we utilized ResNet-50, a widely used architecture.
> > Additionally, to address the concern, we conducted further experiments with architectures such as ResNet-18 and Wide-ResNet. The results of these experiments are provided in Table 15 of the Supplementary Materials, further validating the generalizability of our approach across a range of architectures.
> >
> > We hope our detailed responses have addressed your concerns. We would be pleased to elaborate, if there are any other queries about our work. We would also appreciate it if you could reevaluate the impact of our contributions.
> >
> > ***
> > References:
> >
> > *[7] Saran, Akanksha, et al. "Streaming active learning with deep neural networks." International Conference on Machine Learning. PMLR, 2023.*
> >
> > *[8] Li, Dongyuan, et al. "A Survey on Deep Active Learning: Recent Advances and New Frontiers." IEEE Transactions on Neural Networks and Learning Systems (2024).*

---

> ### Comment · Reviewer_EpRF · 2024-11-26
>
> Thank you for your detailed and thoughtful response to my questions. While some of my concerns have been addressed, several critical issues remain:
>
> - Regarding Q1: Tables 19 and 20 indicate that the training costs of the data selectors are nearly 50% or more of the target model training costs. When combined with the marginal improvements in accuracy, the practical significance of the approach appears limited. Furthermore, comparing the selection costs solely against the baseline and target model training is not entirely fair, as these methods often have higher costs. **It is strongly recommended that the authors compare the actual selection costs against other state-of-the-art (SOTA) methods for a more detailed evaluation.**
>
> - Performance comparison: In Table 1, the main performance comparison only involves the baseline model, which is insufficient to validate the proposed method’s effectiveness. Although the method can integrate scoring mechanisms, **it lacks comparisons with more advanced SOTA methods, which is not convincing.**
>
> - Regarding Q8: Including more advanced deep learning models, such as Vision Transformers (ViT), would further demonstrate the effectiveness of the proposed method.
>
> - The references and methods used for comparison are outdated.
>
> - I have concerns about the generalization of the proposed method across different architectures, i.e., can datasets selected by one trained pruned network generalize well to other networks? This is a crucial issue for data selection methods, as it is impractical to select subsets tailored to every possible model that may be used in the future.
>
> Since some critical concerns are not addressed, I will maintain my score.

---

> > ### Author Response · Authors · 2024-12-02
> >
> > We thank the reviewer for their feedback. We would like to address the concerns in the following responses.
> >
> > >_R1.	Regarding Q1: Tables 19 and 20 indicate that the training costs of the data selectors are nearly 50% or more of the target model training costs. When combined with the marginal improvements in accuracy, the practical significance of the approach appears limited. Furthermore, comparing the selection costs solely against the baseline and target model training is not entirely fair, as these methods often have higher costs. It is strongly recommended that the authors compare the actual selection costs against other state-of-the-art (SOTA) methods for a more detailed evaluation._
> >
> > It is important to note that the reported training time does not fully reflect the benefits of our approach. Although _Tables 19_ and _20_ indicate that the training costs of the data selectors are nearly $50$% as compared to target model training costs in smaller datasets while less than $50$% in large datasets, it is crucial to recognize that the pruned networks require fewer FLOPs and less memory, proportional to the pruning ratio. A detailed comparison is given in _Figure 3_ and _Section A.1_. This reduction in computational complexity can lead to further time savings, especially when similar memory resources as compared to baseline are utilized. Additionally, model fusion leads to faster convergence, which allows us to utilize early stopping to train the target model reducing further $40$% costs. The results of this strategy are provided in _Table 14_, which reduces the overall training time to less than $30$% compared to the baseline. These combined advantages significantly improve the overall training efficiency, making PruneFuse an effective solution compared to baseline methods.
> >
> > We would like to clarify that state-of-the-art techniques as suggested by the reviewer like D2 pruning (ICLR 2024)[1], Moderate (ICLR 2023)[2], and CSS (ICLR 2023)[3] all utilize the pool-based active learning pipeline, which we have considered as the baseline for comparison. Since our contribution is fundamentally orthogonal to these techniques, the costs solely associated with data selection metrics in these techniques are also reflected in PruneFuse. We incorporated these works in our technique in _Table 17_. However, it is essential to note that the primary bottleneck of the pipeline lies in training the data selector network which we aim to optimize. Only a few works, such as SVP[4] and SubSelNet (Neurips 2023)[5], focus on optimizing the entire selection pipeline, and we have provided detailed discussions with these methods in _Section 2, Table 3_ and _Table 13_.
> > We hope this clarifies the prime contribution of this work and how it is orthogonal to the mentioned SOTA techniques.
> >
> > >_R2.	Performance comparison: In Table 1, the main performance comparison only involves the baseline model, which is insufficient to validate the proposed method’s effectiveness. Although the method can integrate scoring mechanisms, it lacks comparisons with more advanced SOTA methods, which is not convincing._
> >
> > To clarify, the baseline in _Table 1_ refers to the pool-based active learning pipeline, which is widely used in data selection methods, including advanced SOTA techniques such as [1], [2], and [3]. These methods, while incorporating different selection metrics, still rely on the standard pool-based framework, which we use as a baseline for comparison in our work. Furthermore, we have compared our framework against techniques like [4] and [5], which focus on optimizing the entire data selection pipeline, and have provided detailed discussions and comparisons with these methods in _Section 2, Table 3_, and _Table 13_.
> >
> > >_R3.	Regarding Q8: Including more advanced deep learning models, such as Vision Transformers (ViT), would further demonstrate the effectiveness of the proposed method._
> >
> > Our current work demonstrates the effectiveness of the data selection pipeline using widely-used models like ResNet-18, ResNet-50, ResNet-56, ResNet-110, ResNet-164 and wide-ResNet. While we had planned to extend this approach to include transformers, such as ViT, due to time constraints, the detailed comparisons are not yet completed. We will include them in the final version of the paper.
> >
> > >_R4.	The references and methods used for comparison are outdated._
> >
> > Please refer to the explanation above in responses R1 and R2.

---

> > > ### Author Response · Authors · 2024-12-02
> > >
> > > >_R5.	I have concerns about the generalization of the proposed method across different architectures, i.e., can datasets selected by one trained pruned network generalize well to other networks? This is a crucial issue for data selection methods, as it is impractical to select subsets tailored to every possible model that may be used in the future._
> > >
> > > Our experiments show that datasets selected by one pruned network generalize well to other architectures, yielding performance comparable to the baseline. However, the accuracy of the target model is slightly lower than PruneFuse. This happens because the data selector no longer benefits from architectural coherence, and the advantages of model fusion are not fully realized. Nevertheless, the selected subsets are based on data representativeness and informativeness, making them applicable across different models. We will include detailed results of these experiments in the revised paper.
> > >
> > > We hope this explanation addresses the reviewer’s concerns.
> > >
> > > ---
> > >
> > > References
> > >
> > > _[1] Maharana, Adyasha, Prateek Yadav, and Mohit Bansal. "D2 pruning: Message passing for balancing diversity and difficulty in data pruning." arXiv preprint arXiv:2310.07931 (2023)._
> > >
> > > _[2] Xia, Xiaobo, et al. "Moderate coreset: A universal method of data selection for real-world data-efficient deep learning." The Eleventh International Conference on Learning Representations. 2022._
> > >
> > > _[3] Zheng, Haizhong, et al. "Coverage-centric coreset selection for high pruning rates." arXiv preprint arXiv:2210.15809 (2022)._
> > >
> > > _[4] Coleman, Cody, et al. "Selection via Proxy: Efficient Data Selection for Deep Learning." International Conference on Learning Representations. 2020._
> > >
> > > _[5] Jain, Eeshaan, et al. "Efficient data subset selection to generalize training across models: transductive and inductive networks." Advances in Neural Information Processing Systems 36 (2023)._

---

> ### Comment · Reviewer_EpRF · 2024-12-03
>
> Dear Authors,
>
> Thank you for your detailed response. I will take the response into account and listen to other reviewers' voices during the AC-reviewer discussion phase and decide whether to raise my rating.
>
> Best regards

---

### Official Review · Reviewer_VjNm · 2024-11-01

**Soundness:** 3
**Presentation:** 3
**Contribution:** 3
**Rating:** 6
**Confidence:** 3

**Summary:**

This paper proposes an approach to accelerate the sample selection process in active learning by leveraging network pruning. To further exploit the power of pruned network, several additional modules, such as network fusion and knowledge distillation, are introduced.

**Strengths:**

- The use of a pruned network as a proxy for selecting informative, unlabeled samples in model training is rational and effective. Although the idea is straightforward, the authors present a well-structured framework incorporating modules such as network fusion for accelerating convergence, knowledge distillation for maintaining performance, and PruneFuse V2 to achieve a favorable accuracy-efficiency balance.
- The paper is well-presented, with clear motivations behind each proposed module and informative visuals, such as Figure 2.

**Weaknesses:**

- In Section 4.1, the pruning process appears to involve removing channels with a low L2 norm from a randomly initialized network. If the network is initialized without training, layers with fewer parameters may naturally have lower L2 norms, resulting in the straightforward removal of those layers. Is this approach widely accepted and theoretically sound? Are there existing studies to support this strategy?
- The motivation behind knowledge distillation is unclear, and its effectiveness is not validated experimentally. An ablation study and further discussion on the role of knowledge distillation in the framework are recommended.
- The performance improvements shown in Tables 1 and 2 are marginal in many cases. Although Params and FLOPS are reduced, the method’s complexity in terms of parameters and operations, such as model fusion and knowledge distillation, raises questions. A direct runtime comparison between the proposed method and baseline methods would be insightful.
- The comparative methods used are somewhat outdated, with BALD and SVP originating from 2019. More recent methods should be included in the comparison.

**Questions:**

- In Algorithm 1, define \( D_j \) in line 6 before using it in line 7.
- Typographical error in Figure 1: “fusiowith.”
- In line 410, clarify the rationale for the unusual training epoch number (181) used for CIFAR.

---

> ### Author Response · Authors · 2024-11-25
>
> We thank the reviewer for the thoughtful feedback on our paper and appreciate the opportunity to address the concerns raised in detail below.
>
> ---
>
> >_1.	Pruning without training leads to straight removal of layers. Are there existing studies to support this strategy._
>
> The approach of pruning from a randomly initialized network, as employed in our work, is well-supported by prior research. Specifically, the Pruning from Scratch [1] demonstrates that effective pruned structures can emerge directly from randomly initialized weights without requiring pretraining. This method broadens the search space for optimal architectures, unlike pruning pre-trained networks, which are inherently biased by their initial training trajectory. Moreover, the Lottery Ticket Hypothesis [2] further supports the feasibility of identifying sparse, trainable subnetworks at initialization.
> In our experiments, pruned models derived from randomly initialized weights consistently demonstrated robust data selection capabilities and effective fusion with the original network. This empirical evidence, combined with the findings of [1] and [2], validates the soundness of our pruning strategy at initialization.
> Additionally, we explore alternatives to the static initial pruning, such as iterative pruning in PruneFuseV2 (Section 4.6), where the trained fused network generates a pruned network for subsequent data selection. Furthermore, we include results for dynamic pruning, where pruning occurs progressively over the first 20 epochs (Table 18 in the Supplementary Materials), to evaluate the impact of different pruning methodologies. These additional experiments demonstrate the flexibility of the proposed framework and its ability to effectively incorporate various pruning strategies.
>
> ---
>
> >_2.	Ablation study on the use of knowledge distillation._
>
> PruneFuse demonstrates strong performance even without incorporating knowledge distillation (KD). However, KD is integrated into the framework to provide additional optimization for the fused model $\theta_F$. By reusing the logits from the pruned model $\theta_p^*$, which are readily available from its training phase, KD is incorporated without incurring any additional computational overhead.
>
> Detailed ablation studies on KD, presented in the Table 10 of the Supplementary Materials, confirm its modest contribution. KD marginally improves performance, particularly in high-label budgets (e.g., $b = 40$%). However, the core performance gains of PruneFuse stem from the proposed model fusion, which significantly enhances both efficiency and convergence.
>
> ---
>
> >_3.	Performance improvements seems marginal and a direct runtime comparison would be insightful._
>
> The primary motivation for our work is to make the time-intensive routine of active learning more efficient, particularly in resource-constrained environments. To support this, we have included detailed runtime comparisons in Table 19 of the Supplementary Materials, which evaluate the computational efficiency of our method across various architectures and datasets. Our results demonstrate that PruneFuse achieves substantial reductions in computational overhead compared to baseline methods.
>
> ---
>
> >_4.	Comparison with recent works._
>
> While we recognize the value of recent works, we specifically chose SVP [15] as our primary baseline due to its direct comparability and versatility.  We reference many recent works in Section 2, including SubSelNet (NeurIPS 2023) [3]; however, SubSelNet requires a computationally intensive pre-training routine on a large pool of architectures and this process must be repeated for any change in data or model distribution. Such demands can be impractical, particularly in resource-constrained or dynamic environments. In contrast, SVP is a more practical and effective benchmark for data selection, which is why we chose it for comparison.
>
> To provide a broader evaluation, we also implemented three recent coreset selection techniques, Forgetting-events [4], Moderate [5] and CSS [6], and present a detailed comparison in Table 17 of the Supplementary Materials. These results demonstrate that PruneFuse remains effective when combined with these advanced scoring techniques, maintaining its computational efficiency while achieving strong data selection performance across diverse scenarios. This further establishes PruneFuse as a robust and adaptable framework for active learning.

---

> > ### Author Response · Authors · 2024-11-25
> >
> > >_5.	Clear the rational of using 181 epoch number and fixing typos._
> >
> > Thank you for pointing out the typos and unclear details. We have addressed these issues in the revised version of the paper. Regarding the specific choice of 181 epochs, this follows the experimental setup used by SVP [7], ensuring a fair and consistent basis for comparison.
> >
> >
> > We believe we clarified all your concerns. Should you have any further questions or require additional clarifications, we would be pleased to provide them. We would also be grateful if you could reevaluate the significance of this work in the light of revisions provided.
> >
> > ***
> >
> > References:
> >
> > *[1] Wang, Yulong, et al. "Pruning from scratch." Proceedings of the AAAI conference on artificial intelligence. Vol. 34. No. 07. 2020.*
> >
> > *[2] Frankle, Jonathan, and Michael Carbin. "The Lottery Ticket Hypothesis: Finding Sparse, Trainable Neural Networks." International Conference on Learning Representations. 2018.*
> >
> > *[3] Jain, Eeshaan, et al. "Efficient data subset selection to generalize training across models: transductive and inductive networks." Advances in Neural Information Processing Systems 36 (2023).*
> >
> > *[4] Toneva, Mariya, et al. "An Empirical Study of Example Forgetting during Deep Neural Network Learning." International Conference on Learning Representations. 2018.*
> >
> > *[5] Xia, Xiaobo, et al. "Moderate coreset: A universal method of data selection for real-world data-efficient deep learning." The Eleventh International Conference on Learning Representations. 2022.*
> >
> > *[6] Zheng, Haizhong, et al. "Coverage-centric Coreset Selection for High Pruning Rates." The Eleventh International Conference on Learning Representations. 2023.*
> >
> > *[7] Coleman, Cody, et al. "Selection via Proxy: Efficient Data Selection for Deep Learning." International Conference on Learning Representations. 2020.*

---

### Meta-Review · Area_Chair_SiFu · 2024-12-23

**Metareview:**

The submission proposes PruneFuse - using pruned networks to select informative data subsets to train the original model for the highest accuracy. The pruned model can be efficiently run on a larger number of input samples, and then trained on a selected subset. The now trained pruned model can be fused back into the original larger model, and the fused larger model can be trained further. The authors show that this method can accelerate the overall training process and achieve better accuracy than baselines.

The submission originally received ratings of 6, 5, 5, which were downgraded to 6, 3, 5, ultimatey leaning negative in aggregate.
The reviewers outlined multiple weaknesses including:
1) Lack of comparison with prior work in the area of coreset selection that is relevant and very related.
2) Lack of theoretical background or extensive analysis of the relationship between pruning ratio, final accuracy, number of samples, total FLOPs, etc.

Ultimately the key contribution of this work is that a pruned model can be used as proxy for the larger original model for selecting relevant samples in an active learning setting. Based on discussions with reviewers, this alone does not meet the bar for acceptance.

The ACs do not find sufficient reason to overturn the negative consensus and choose to reject the submission.

**Additional Comments On Reviewer Discussion:**

In the discussion with reviewers, the reviewers reiterated the lack of comprehensive comparisons with related state-of-the-art approaches, both in terms of accuracy and training costs. The added complexity of the method, and the resulting marginal improvements bring into question the efficacy of this method.

---

### Decision · Program_Chairs · 2025-01-22

Reject